# Conformal C2ST: Turning weak classifiers into strong two-sample tests

**Vansh Bansal** [* 1]   **Tianyu Chen** [* 1]   **James G. Scott** [1]

## Abstract

The two-sample testing problem, a fundamental task in statistics and machine learning, seeks to determine whether two sets of samples, drawn from underlying distributions $p$ and $q$, are in fact identically distributed (i.e. whether $p = q$). A popular and intuitive approach is the classifier two-sample test (C2ST), where a classifier is trained to distinguish between samples from $p$ and $q$. Yet despite simplicity of the C2ST, its reliability hinges on access to a near-Bayes-optimal classifier, a requirement that is rarely met and difficult to verify. This raises a major open question: can a weak classifier still be useful for two-sample testing? We show that the answer is a definitive yes. Building on the work of Hu & Lei (2024), we analyze two conformal variants of the C2ST that convert the scores from any trained classifier—even if weak, biased, or overfit—into exact, finite-sample p-values. We establish two key theoretical properties of the conformal C2ST: (i) finite-sample Type-I error control, and (ii) non-trivial power that degrades gently in tandem with the error of the trained classifier. The upshot is that even poorly performing classifiers can yield powerful and reliable two-sample tests. This general framework finds a powerful application in Bayesian inference, particularly for validating Neural Posterior Estimation (NPE) models, where the task of comparing a learned posterior approximation $q(\theta \mid y)$ to the true posterior $p(\theta \mid y)$ can be framed as a two-sample test. Empirically, the Conformal C2ST outperforms classical discriminative tests across a wide range of benchmarks for this task. Our results establish the conformal C2ST as a practical, theoretically grounded diagnostic tool. The code is available at https://github.com/TianyuCodings/conformal_c2st.

## 1 Introduction

A fundamental problem in statistics and machine learning is to assess whether two sets of samples, drawn from distributions $p(x)$ and $q(x)$, are in fact distributed identically. To this end, two-sample tests (Lehmann & Romano, 2005) summarize the differences between the samples into a test statistic, which is then used to test the null hypothesis that $p = q$. A key application for modern two-sample tests is evaluating the sample quality of generative models, where $p$ represents the true data-generating process and $q$ is a trained neural approximation.

A popular tool for evaluating generative models is the classifier-based two-sample test (C2ST) (Lopez-Paz & Oquab, 2016), a flexible, general-purpose approach. The C2ST reframes the problem of distributional comparison as a classification task. In this approach, a classifier is trained to distinguish between samples drawn from the true distribution $p(x)$, versus those drawn from the neural approximation $q(x)$. If the classifier achieves high accuracy, it suggests that the samples from $p$ and $q$ are distinguishable, indicating a mismatch between the two distributions. Conversely, poor classification performance implies that the two sets of samples are statistically similar, supporting the validity of the approximation.

Yet while C2ST is valid under the null—i.e., it controls type-I error asymptotically—its power depends critically on the quality of the classifier, which has the difficult task of learning a global decision boundary over the space $\mathcal{X}$. Prior theoretical works in this direction also assume access to a Bayes-optimal or a powerful asymptotically consistent classifier (Kim et al., 2020; 2019). This problem persists even in more advanced classifier-based methods like Discriminative Calibration (DC) (Yao & Domke, 2023), $\ell$-C2ST (Linhart et al., 2023) and AutoML (Kübler et al., 2023), which also rely on access to a near-optimal classifier. In practice, however, a trained classifier may be weak due to limited training data, restricted model capacity, poor calibration of output probabilities (e.g. thresholding at 0.5), or simply the inherent challenge of the classification problem. This raises a potential ambiguity in interpretation: if a model $q$ "passes" the C2ST, that could mean either that $q = p$, or that $q \neq p$ but the classifier is too weak or poorly trained to tell the difference.

---

[*]Equal contribution  [1]Department of Statistics and Data Sciences, University of Texas at Austin, United States. Correspondence to: Vansh Bansal <vansh@utexas.edu>.

*Proceedings of the 43rd International Conference on Machine Learning*, Seoul, South Korea. PMLR 306, 2026. Copyright 2026 by the author(s).

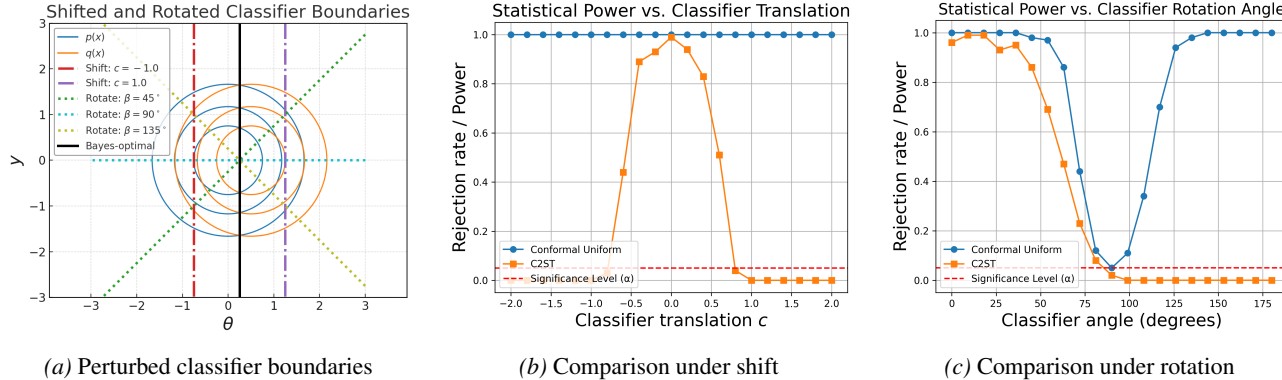

*(a)* Perturbed classifier boundaries     *(b)* Comparison under shift     *(c)* Comparison under rotation

*Figure 1.* Power of the C2ST and conformal C2ST under shift and rotate perturbations of the optimal decision boundary. The conformal test is much more robust to a weak or misspecified classifier.

This evaluation challenge is particularly acute in Neural Posterior Estimation (NPE), an increasingly popular and practical tool for Bayesian inference. NPE methods use simulations to train a deep generative model $q(\theta \mid y)$ to approximate the true, typically intractable posterior $p(\theta \mid y)$ (Ho et al., 2020; Geffner et al., 2023; Chen et al., 2025; Gloeckler et al., 2024; Papamakarios et al., 2021; Wildberger et al., 2023; Kingma et al., 2013). While powerful, this approach creates a critical validation problem: how can we verify whether the learned posterior $q$ faithfully approximates the true posterior $p$? We restrict our focus to this $\mathcal{M}$-closed validation of the estimate $q$, distinguishing it from the separate $\mathcal{M}$-open challenge of checking if the simulator $p$ matches the real-world data generating process. Several diagnostics exist for this purpose, yet they all suffer from drawbacks. Simulation-based calibration (SBC) (Talts et al., 2018) is a widely used tool. Yet in its original form (rank-SBC), it operates only on one-dimensional marginals. This not only creates a potentially severe multiple-testing problem, but also leaves rank-SBC insensitive to inaccuracies that affect the joint distribution of all parameters without affecting their one-dimensional marginals. Proposed fixes that use the joint likelihood as a test statistic (Modrák et al., 2023) are inapplicable to many NPE problems, where likelihoods are intractable. Another recent method called Test of Accuracy with Random Points (TARP) (Lemos et al., 2023) is highly sensitive to the specification of a non-trainable proposal distribution, limiting its practical use.

**Summary of contributions.** We address these limitations by introducing the Conformal C2ST. Our approach utilizes the conformal framework of Hu & Lei (2024), who use the oracle density ratios $p/q$ to build two-sample tests for conditional distributions with unequal $y-$marginals. We, on the other hand, concentrate on the equal marginal conditional or unconditional regime for evaluating accessible generative models– if present, the true conditioning input can be given to the model unlike the case when only samples are

available from both. Moreover, since density ratios are only a monotone transform of the *Bayes-optimal* classifier, our tests appear as a special case of theirs, however our motivations are orthogonal. Hu & Lei (2024) rely on access to a strong classifier to model the density ratios which often fails in practice. This is further problematic in settings like NPE, where classifiers are often trained on limited data and may be too weak to find an optimal decision boundary.

We instead target the realistic, imperfect regime. By providing a rigorous robustness theory for weak plug-in classifiers, we show how conformal calibration can transform a weak, misspecified, or overfit classifier into a powerful and trustworthy two-sample test. Specifically, we first show that conformal calibration effectively decouples type-I error control from the classifier's discriminative accuracy. More importantly, we also prove that our conformal tests maintain meaningful power when $p \neq q$, even when the classifier is weak or poorly trained. Intuitively, they do so by aggregating weak but informative ranking signals, a property that is crucial in scenarios where the traditional C2ST struggles, such as high-dimensional densities, small-sample regimes, and low signal-to-noise ratio tasks.

Crucially, we provide two conformal tests and show that rank-based post-processing offers a flexible computational trade-off: The uniform test maximizes power through resampling fresh calibration sets, while the multiple test *strictly matches* the simulation budget of the standard C2ST while still being more powerful and robust to weak classifiers. We support these theoretical developments with extensive empirical results, showing that both variants of the Conformal C2ST exhibit state of the art performance across a wide range of two-sample testing problems.

**A toy example.** Before detailing our results, we briefly focus on a toy example, to illustrate the fundamental advantage of conformal calibration. The setup mimics the NPE setting, where the marginal distribution of $y$ is preserved under both

the true and approximate joint distributions. Specifically, we consider two bivariate normal distributions: $p(\theta, y)$ is a standard bivariate normal, $\mathcal{N}(\mathbf{0}, I_2)$, while $q(\theta, y)$ is the same distribution with a mean shift of $0.5$ in the $\theta$ coordinate, i.e., $\mathcal{N}([0.5, 0]^\top, I_2)$. The Bayes-optimal classifier for distinguishing between samples from $p$ and $q$ is a linear decision boundary at $\theta = 0.25$, bisecting $\mathbb{R}^2$ between the two means. Classifier scores are obtained by computing the signed distance from a sample point $(\theta, y)$ to the decision boundary.

We then degrade the classifier by manipulating the decision boundary in two systematic ways. First is *translation*, where the decision boundary is shifted horizontally by a parameter $c$, modifying the decision boundary to $\theta = 0.25 + c$. The parameter $c$ can be positive, shifting the boundary toward $q$, or negative, shifting it toward $p$. As $|c|$ increases, the boundary moves further from its optimal position, increasing classification error. As we see in Figure 1b, this rapidly degrades the power of the C2ST for distinguishing $p$ and $q$. Yet the conformal C2ST's performance remains remarkably stable, showing no degradation in power even for large shifts. This suggests a strong level of robustness to biased or poorly calibrated classifiers. (Although the classifier is trained to distinguish individual draws from $p$ and $q$, the goal of neural posterior testing is to assess whether two *collections* of samples arise from the same distribution, allowing aggregation across multiple draws and yielding greater power than would be expected from single-point classification alone.) The second form of degradation is *rotation*, where the decision boundary is rotated about the midpoint between the means of $p$ and $q$. The boundary under rotation is described by $(\theta - 0.25)\cos\beta + y\sin\beta = 0$, where $\beta$ controls the angle of rotation. A rotation of $\beta = 0$ corresponds to the optimal linear discriminant, while increasing $|\beta|$ introduces growing misalignment. Figure 1c shows that the conformal C2ST continues to perform significantly better under quite severe rotations, indicating resilience even when the classifier is badly misaligned. At $\beta = \pi/2$, the decision boundary becomes completely orthogonal to the separation axis of the two distributions, rendering the distribution of classifier scores identical under $p$ and $q$. In this extreme case, the conformal C2ST has power $0.05$, which is precisely the type-I error rate of the test.

These results illustrate the key advantage of the conformal C2ST: even when the classifier itself becomes progressively poorer or more miscalibrated, its ranking signals nonetheless remain useful.

## 2 Preliminaries

**Notation.** While the conformal C2ST is general two-sample test, we focus on the NPE setting as a motivating problem. Consider two distributions over joint parame-

ter–observation pairs $(\theta, y) \in \Theta \times \mathcal{Y}$: the true posterior $p(\theta \mid y)$, defined via the joint density $p(\theta, y) = \pi(\theta)p(y \mid \theta)$, and an approximate posterior $q(\theta \mid y)$, learned using simulation-based inference (SBI). We are agnostic as to *how* $q$ was learned; instead, our goal is to assess whether $q(\theta \mid y) = p(\theta \mid y)$ almost surely over $\Theta \times \mathcal{Y}$. Assuming a shared marginal $\pi(y)$, we define the approximate joint as $q(\theta, y) := \pi(y)q(\theta \mid y)$. Under this assumption, testing equality of posteriors reduces to testing the null hypothesis $H_0 : p(\theta, y) = q(\theta, y)$ almost surely over $\Theta \times \mathcal{Y}$. This naturally frames the problem as a *two-sample test* between the joint distributions $p(\theta, y)$ and $q(\theta, y)$.

In the NPE setting, i.i.d. samples from both $p$ and $q$ are readily available. To sample from $p$, we draw $\theta \sim \pi(\theta)$ and then $y \sim p(y \mid \theta)$. To sample from $q$, we use the $y$ margin only of the true $p$, implicitly generating $y \sim \pi(y)$, and then we sample $\theta \sim q(\theta \mid y)$ from the learned model. For convenience, we write $x = (\theta, y)$, and use $p$ and $q$ as shorthand for the true and approximate joint densities over $x$. We denote samples as $\{X_i\}_{i=1}^n \sim p$ and $\{\tilde{X}_i\}_{i=1}^n \sim q$. Throughout, we assume that $p(\theta \mid y)$ and $q(\theta \mid y)$ are absolutely continuous with respect to a common base measure.

**The classifier two-sample test (C2ST).** The C2ST is a widely used approach for detecting differences between two distributions. In the context of posterior validation, suppose we are given two datasets $\{x_i\}_{i=1}^n \sim p$ and $\{\tilde{x}_i\}_{i=1}^n \sim q$. To test the null hypothesis $H_0 : p = q$, a classifier is trained to distinguish between the two samples. Specifically, each $x_i$ is assigned label 1 and each $\tilde{x}_i$ is assigned label 0, producing a labeled dataset $\{(z_i, \ell_i)\}_{i=1}^{2n}$, where $z_i \in \mathcal{X}$ and $\ell_i \in \{0, 1\}$. The combined dataset is randomly split into disjoint training and testing subsets. A classifier $f : \mathcal{X} \to [0, 1]$ is trained on the training portion to estimate the conditional probability $\mathbb{P}[\ell = 1 \mid z]$, and evaluated on the test set $\mathcal{D}_{\text{te}}$. The C2ST test statistic is the classification accuracy on the test set:

$$\hat{t} := \frac{1}{n_{\text{te}}} \sum_{(z_i, \ell_i) \in \mathcal{D}_{\text{te}}} \mathbb{I}\{\mathbb{I}\{f(z_i) > 1/2\} = \ell_i\},$$

which is shown to have an asymptotic normal distribution by Lopez-Paz & Oquab (2016). The null hypothesis $H_0 : p = q$ is rejected if $\hat{t}$ is significantly greater than 0.5, indicating that the classifier has learned to distinguish between $p$ and $q$.

While C2ST is easy to implement and effective in many cases, it has two key limitations. First, its power depends heavily on the classifier quality. Poorly trained or underfit classifiers can yield inconclusive results, even when the two distributions differ. Second, its test statistic relies on hard decisions via thresholding at 0.5, discarding potentially useful information about classifier confidence.

# 3 The uniform test: conformal calibration of classifier scores

To address these limitations, we adopt a conformal framework that calibrates classifier scores directly, rather than thresholding them. This yields a method we term the *Conformal C2ST*. The core idea is to treat each point as a test case, compute a score for that point (e.g., classifier log-odds), and use a conformal p-value to assess how extreme that score is relative to a calibration sample from the reference distribution ($p$). Each such p-value reflects the plausibility of a single test point from $q$ under the null. To test whether $p = q$ globally, we repeat this procedure across many independent draws from $q$, aggregating the resulting p-values to assess overall deviation from uniformity. This allows the method to extract and accumulate weak signals for assessing distributional equality, even from underperforming classifiers, by calibrating based on ranks rather than raw accuracy.

Concretely, let $\{X_1, \ldots, X_m\} \sim p$ be a calibration sample and let $\tilde{X} \sim q$ be a test point. Let $s : \mathcal{X} \to \mathbb{R}$ be a deterministic scoring function, which in our case will be the output of a classifier trained to distinguish $p$ from $q$, just like in C2ST. (Below we discuss the specific choice of $s$.) Define the nonconformity scores $S_i = s(X_i)$ for $i = 1, \ldots, m$, and $S_{m+1} = \tilde{S} = s(\tilde{X})$. The conformal p-value (Vovk et al., 2005; Lei & G'Sell, 2018) for $\tilde{X}$ is then given by:

$$U := \frac{\sum_{i=1}^{m+1} \mathbb{I}\left\{S_i < \tilde{S}\right\} + \xi \cdot \sum_{i=1}^{m+1} \mathbb{I}\left\{S_i = \tilde{S}\right\}}{m + 1} \quad (1)$$

where $\xi \sim \text{Unif}[0, 1]$ is a tie-breaking random variable. The p-value $U$ reflects the relative rank of the test point's score among the calibration scores.

## 3.1 Results on validity and power

A key advantage of this approach is that it inherits the finite-sample validity guarantee from conformal prediction. Under $H_0$, the calibration points and the test point are exchangeable, implying the following marginal guarantee.

**Lemma 3.1** (Uniformity under the null). *Under the null hypothesis $H_0 : p = q$, the conformal p-value $U$ defined in* (1) *satisfies:*

$$\mathbb{P}(U \leq u \mid H_0) = u, \quad \forall u \in [0, 1],$$

*where the probability is over the random draw of the calibration sample, the test point, and the tie-breaking variable.*

This result holds for any deterministic scoring function $s$ and for any finite calibration size $m$, making the conformal C2ST robust to classifier quality and sample size. This result is a standard property of conformal inference (Vovk et al., 2005; Lei & G'Sell, 2018).

**From marginal p-values to a uniformity test.** To turn the marginal validity established in Lemma 3.1 into a two-sample test of $H_0 : p = q$, we repeat the conformal p-value computation across multiple test points. Specifically, let $\{\tilde{X}_j\}_{j=1}^{n_q} \sim q$ be independent test samples. For each $j$, we draw an independent calibration set $\mathcal{C}_j = \{X_{j,i}\}_{i=1}^m \sim p$. We compute the scores $S_{j,i} = s(X_{j,i})$, and $\tilde{S}_j = s(\tilde{X}_j)$ and the corresponding conformal p-value as

$$U_j := \frac{\sum_{i=1}^{m+1} \mathbb{I}\left\{S_{j,i} < \tilde{S}_j\right\} + \xi_j \cdot \sum_{i=1}^{m+1} \mathbb{I}\left\{S_{j,i} = \tilde{S}_j\right\}}{m + 1},$$

where $S_{j,m+1} = \tilde{S}_j$, and $\xi_j \sim \text{Uniform}[0, 1]$ are independent tie-breaking variables. Under $H_0$, each $U_j \sim \text{Unif}[0, 1]$, so we can aggregate the $\{U_j\}$ to form a global test statistic. We consider the empirical CDF $\hat{G}(u) = \frac{1}{n_q} \sum_{j=1}^{n_q} \mathbb{I}\{U_j \leq u\}$ and apply the one-sample Kolmogorov–Smirnov (KS) test[1] (Lehmann & Romano, 2005), using

$$T_{\text{KS}} = \sup_{u \in [0,1]} \left|\hat{G}(u) - u\right|.$$

Because $H_0 : p = q$ implies exchangeability between the calibration and test samples, each $U_j$ is uniformly distributed, and the KS test controls Type-I error at level $\alpha$. (We refer to this in the benchmarks as the "uniform" test.)

**Power.** While Lemma 3.1 ensures exact marginal validity of conformal p-values for *any* deterministic score function $s$, the power of the test under the alternative $H_1 : p \neq q$ crucially depends on how well $s$ separates samples from $p$ and $q$. A natural and theoretically grounded choice is the *oracle density ratio* between the joint distributions (or equivalently their conditionals for $\theta$, since $p(y) = q(y)$). Let $r(x) = p(x)/q(x)$ denote this true density ratio. If $\eta(x) := \mathbb{P}(l = 1 \mid x)$ is the output of the Bayes classifier distinguishing between $x \sim p$ (label $l = 1$) and $x \sim q$ (label $l = 0$), then the density ratio is related to classifier scores via:

$$r(x) = \frac{\eta(x)}{1 - \eta(x)}. \quad (2)$$

We will soon consider what happens when $r$ is estimated with error, but for now we suppose it is known. Because the transformation $t \mapsto t/(1 - t)$ is strictly increasing on $(0, 1)$, the rankings induced by $r(x)$ and the probabilities $\eta(x)$ are identical.

These rankings are of central importance to the conformal method, which depends only on the orderings of scores, not their magnitudes. A natural way to quantify the quality

---

[1] We utilize the KS test for its interpretability and established use in the literature, though our framework is compatible with any valid uniformity test (e.g., Anderson-Darling, Cramer-von Mises).

of such a ranking is the area under the ROC curve (AUC) (Fawcett, 2006), which measures how well a scoring function separates the two distributions. Specifically, the AUC for $r(x)$ is given by:

$$\text{AUC}(r) = \mathbb{P}\left[r(X) > r(\tilde{X})\right] + \frac{1}{2}\mathbb{P}\left[r(X) = r(\tilde{X})\right],$$

which reflects the probability that a randomly chosen sample $X \sim p$ is ranked above $\tilde{X} \sim q$ by the scoring function $r(x)$. The next lemma formalizes that $r(x)$ (or any monotonic transformation of it) maximizes this quantity.

**Lemma 3.2.** *For any measurable scoring function $s : \mathcal{X} \to \mathbb{R}$, $\text{AUC}(s) \le \text{AUC}(r)$, with equality if and only if there exists a strictly increasing function $h$ such that $s(x) = h(r(x))$ for $q$-almost every $x$.*

This result justifies the use of the density ratio $r(x)$ as an optimal scoring function for discriminating between $p$ and $q$; see Appendix A.1 for the full statement and proof.

Moreover, AUC serves as a useful proxy for the power of the conformal uniformity test, because it quantifies how well the scoring function ranks samples from $p$ above those from $q$. Specifically, as the number of calibration samples $m \to \infty$, the conformal p-value for a test point $\tilde{X} \sim q$ converges almost surely to its population-level limit:

$$U \xrightarrow[m\to\infty]{\text{a.s.}} \mathbb{P}_{X\sim p}\left[r(X) < r(\tilde{X})\right] + \xi \mathbb{P}_{X\sim p}\left[r(X) = r(\tilde{X})\right],$$

where $\xi \sim \text{Unif}(0,1)$. When $r(x)$ has good separation between the distributions, these p-values tend to concentrate below 0.5, making them detectably non-uniform.

As the number of test points $n_q \to \infty$, the empirical CDF $\hat{G}(u) = \frac{1}{n_q}\sum_{j=1}^{n_q}\mathbb{I}\{U_j \le u\}$ converges to the population CDF $G(u) = \mathbb{P}(U \le u)$, and the Kolmogorov–Smirnov test statistic converges to $\sup_{u\in[0,1]}|G(u) - u|$, which captures the deviation from uniformity. When AUC exceeds 0.5, i.e., $r(x)$ tends to rank $p$ above $q$, conformal p-values become stochastically smaller than uniform. This deviation drives up the KS statistic and hence the test power. Moreover, the expected conformal p-value under the alternative is directly related to AUC, as the following lemma establishes.

**Lemma 3.3.** *Under the alternative $H_1 : p \ne q$, we have*

$$\mathbb{E}[U] = 1 - \text{AUC}(r) \le \frac{1}{2} - \frac{TV(p,q)}{2} < \frac{1}{2}$$

*in the asymptotic limit of $m \to \infty$.*

This result shows that, under the alternative, conformal p-values skew toward zero, with the extent of skewness proportional to the total variation distance between the two distributions. (Hu & Lei, 2024) also motivate the density

ratio $r(x) = p(x)/q(x)$ from an information-theoretic perspective, showing that its variability under $q$ controls the deviation from uniformity; see Lemma A.2 in the Appendix.

## 3.2 Robustness to weak classifiers

Our above analysis involves the oracle density ratio $r(x)$. But in practice, the density ratio is often approximated using a classifier trained to distinguish between the true and approximate joint distributions. Accordingly, our analysis explicitly incorporates an error-prone plug-in estimate $\hat{r}$, derived from a potentially weak classifier $\hat{\eta}$ in (2).

Our main theoretical result shows that, under mild regularity conditions on the true density ratio, the uniform conformal test retains high power, even when the scoring function is imperfect. Our result specifically relates the error of the estimated density ratio to the performance of the conformal C2ST.

**Assumption 3.4** (Low noise condition). The random variable $Z := r(\tilde{X}) - r(\tilde{X}')$, where $\tilde{X}, \tilde{X}' \overset{\text{i.i.d.}}{\sim} q$, has a density that is bounded in a neighborhood around zero.

We emphasize that Assumption 3.4 is a mild regularity condition typically satisfied in practice. Provided $p$ and $q$ have overlapping support and the true density ratio $r(x)$ is relatively smooth, the difference $Z = r(\tilde{X}) - r(\tilde{X}')$ admits a finite density around zero. This precludes degenerate scenarios where $r(x)$ is locally constant. Concretely, Assumption 3.4 fails only when the oracle ratio carries almost no ranking signal — that is, when too many pairs $(\tilde{X}, \tilde{X}')$ are nearly tied under $r$. A natural failure case is when both $p$ and $q$ are very flat over broad regions, so that $r$ varies little and most pairwise comparisons become near-indistinguishable; in this regime the alternative is intrinsically hard for *any* ranking-based test to detect. Theoretically, this requirement corresponds to a pairwise Tsybakov-type margin condition (Audibert & Tsybakov, 2007) with noise exponent $\alpha = 1$. This is strictly weaker than the uniform noise assumption employed by Clémençon et al. (2006, Section 5.1) or the requirement that the density of $r$ itself be uniformly bounded, found in Hu & Lei (2024).

**Theorem 3.5** (Robustness to estimation error). *Under Assumption 3.4, let $\hat{U}$ denote the conformal p-value computed using $m$ calibration points per test point and the approximate score function $\hat{r}$ whose estimation error is given by $\mathbb{E}_{\tilde{X}\sim q}\left[(\hat{r}(\tilde{X}) - r(\tilde{X}))^2\right] \le \varepsilon^2$. Then under the alternative $H_1 : p \ne q$, there exists $M > 0$, depending on $\varepsilon$, $p$, and $q$, such that*

$$\mathbb{E}[\hat{U}] - \mathbb{E}[U] \le \mathcal{O}(\varepsilon^{2/3}) \quad \text{for all } m > M.$$

This theorem shows that the uniform test enjoys a remarkable degree of robustness: even when the classifier is weak

or undertrained, the resulting p-values still exhibit systematic deviation from uniformity under the alternative. In particular, the expected p-value computed using an approximate score $\hat{r}$ remains close to the ideal value obtained from the oracle score $r$, with the error controlled by the quality of the approximation. As long as the estimated score preserves a reasonable approximation to the true ranking, the test maintains power. This highlights a key advantage of the conformal approach: it leverages relative score orderings rather than relying on absolute classifier accuracy; We defer the proof to Appendix A.2. We further show in Lemma A.3 in the Appendix, assuming zero tie probability for continuous classifier outputs, the variance of the conformal p-values scales as $\mathcal{O}(1/m)$. Consequently, the p-values converge quickly to their true non-uniform values under the alternative as one increases $m$, leading to a better test power.

## 4 The multiple test

A potential limitation of the uniform test described above is the need to generate a fresh calibration set for each test point. In many NPE settings, this is not a big issue; drawing from the true joint distribution $p(\theta, y)$ will often be cheap, as it does not require forward simulation through the generative model. For scenarios where sampling from $p(\theta, y)$ is computationally expensive (e.g., high-fidelity physical simulators), we analyze the Conformal Multiple test. This variant utilizes a single shared calibration set, matching the inference cost of the standard C2ST while still retaining higher power and robustness under classifier degradation as we show shortly.

To correct for the dependence introduced by using a shared calibration set, Hu & Lei (2024) propose the average of the conformal p-values computed using a single calibration set as a two-sample U-statistic. They derive an asymptotically valid test statistic by normalizing the deviation of the average p-value from $1/2$, which is the expected value under the null. In our setting, we can simplify their test statistic to obtain:

$$\hat{T} = \frac{\frac{1}{2} - \frac{1}{n_q}\sum_{j=1}^{n_q}\hat{U}_j}{\hat{\sigma}/\sqrt{n_p}} \tag{3}$$

where $\hat{U}_j := \frac{1}{n_p}\sum_{i=1}^{n_p} h_{\hat{r}}(X_i, \tilde{X}_j; \xi_j)$ with the kernel $h_s(x, y; \xi) := \mathbb{I}\{s(x) < s(y)\} + \xi\mathbb{I}\{s(x) = s(y)\}$. Note that $\hat{U}_j$ is obtained from the entire calibration set (common to all test points) with $\hat{r}(\cdot)$ as the scoring function, and $\hat{\sigma}$ is the asymptotic estimated standard deviation of $\frac{\sqrt{n_p}}{n_q}\sum_{j=1}^{n_q}\hat{U}_j$; see Appendix A.4 for details. Under the null, $\hat{T} \sim \mathcal{N}(0, 1)$, whereas under the alternative, $\hat{T} \to \infty$ asymptotically under the assumptions mentioned by Hu & Lei (2024, Theorem 2). We now give our second main result highlighting the robustness of the multiple test, which

directly inherits our analysis from the uniform test. Before proceeding, we make the following tie assumption.

**Assumption 4.1** (Rare ties condition). The score $\hat{r}$ is almost surely continuous (so ties have zero probability), or ties occur at rate $\mathcal{O}_p(1)$.

Note that Assumption 4.1 is mild, and permits a wide range of practical scenarios in which $\hat{r}(x)$ is estimated via smooth classifier outputs. It can fail when the learned score $\hat{r}$ is heavily discretized, quantized, clipped, or nearly constant, since such scores produce a non-negligible mass of exact ties that violate the rate-$O_p(1)$ condition.

**Theorem 4.2.** *Denote by $\hat{T}(s)$ the test statistic obtained using the score function $s$. Further define $\Delta_s := AUC(s) - \frac{1}{2}$, and let Assumption 4.1 and the same conditions as Theorem 3.5 hold. Then under the alternative, i.e., $\Delta_r > 0$, for large $n_p, n_q$, we have that*

*(i) the oracle statistic scales as $\hat{T}(r) = \mathcal{O}_p(\sqrt{\min\{n_p, n_q\}}\Delta_r)$.*

*(ii) the relative oracle-estimate gap scales as:*

$$\frac{\hat{T}(\hat{r}) - \hat{T}(r)}{\hat{T}(r)} = \mathcal{O}_p(\epsilon^{2/3})$$

This theorem again highlights the remarkable robustness obtained by the simple rank-based post–processing used by our conformal statistic. The result shows that, under the same sample budget, the oracle–estimate gap in our conformal two sample U-statistic grows much slower than the oracle statistic itself, retaining near oracle power. Consequently, unlike ordinary C2ST, even when $\hat{r}$ is imperfect, the degradation in the multiple test's power is governed by *misranking*, rather than inaccurate probability (or ratio) estimation.

## 5 Experiments

We now empirically evaluate the performance of the conformal C2ST against baseline methods in a scenario with controlled perturbations of a known reference distribution. The experiments are designed to address two key questions. The first concerns *power under proper training:* when the classifier is well trained, how does the conformal C2ST compare to competing methods? The second concerns *robustness to classifier degradation:* as we progressively degrade the quality of the classifier, does the conformal C2ST retain power better than the ordinary C2ST?

### 5.1 Controlled posterior perturbations.

Our first set of experiments involve a controlled simulation environment with a known ground-truth $p(\theta \mid y)$. From this

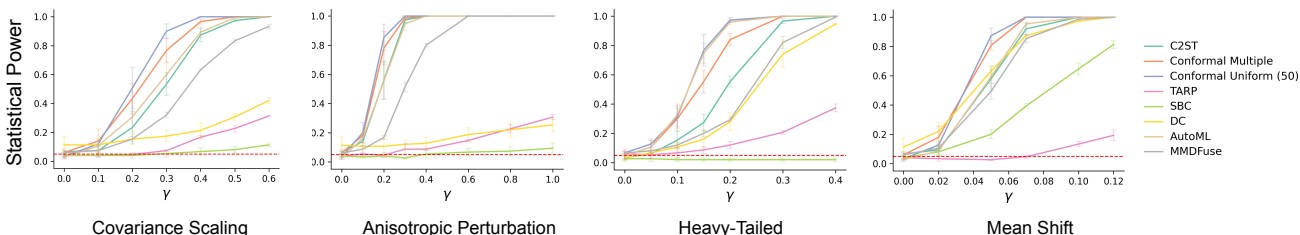

*(a)* Power curves as a function of perturbation level $\gamma$ at fixed classifier quality ($\beta = 0$).

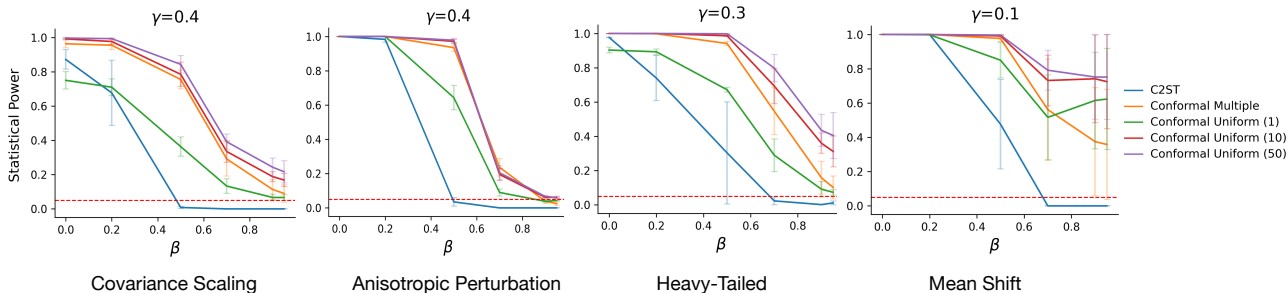

*(b)* Power curves as a function of classifier degradation level $\beta$ at fixed perturbation strength $\gamma$.

*Figure 2.* Statistical power of C2ST and conformal variants across benchmark perturbations. Panel (a) evaluates sensitivity to posterior mismatch; panel (b) evaluates robustness to classifier degradation.

ground truth, we generate a series of flawed approximations $q(\theta \mid y)$, by systematically applying a controlled perturbation of $p$. The magnitude of the perturbation is controlled by a scalar $\gamma$, which acts as a "difficulty dial." When $\gamma = 0$, the approximation is perfect $q = p$, allowing us to assess a method's Type-I error rate. As $\gamma$ increases, the approximation becomes progressively worse, allowing us to measure a test's power.

Our perturbations, described in detail in Appendix B, are designed to mimic common failure modes in NPE, such as biased means, overdispersion, miscalibrated covariance structure, or mode collapse. This framework allows us to directly assess whether a testing method can reliably detect meaningful discrepancies in a setting that mirrors real-world validation challenges. Specifically, we include four perturbation types from NPTBench (Chen et al., 2024) in the main text and defer the rest to Appendix B.

**Testing classifier degradation.** After training a classifier on a given benchmark problem at a fixed perturbation level $\gamma$, we generate a family of degraded classifiers by linearly interpolating the trained model parameters with those of a randomly initialized model:

$$\psi_\beta := (1 - \beta) \cdot \psi_{\hat{\eta}} + \beta \cdot \psi_{\text{rand}}, \quad \beta \in [0, 1],$$

where $\psi_{\hat{\eta}}$ and $\psi_{\text{rand}}$ are the parameter vectors of the trained and randomly initialized classifiers, respectively. This setup allows us to systematically degrade the classifier's quality by varying $\beta$, from a fully trained model ($\beta = 0$) to a

random, uninformative one ($\beta = 1$). Class probabilities and nonconformity scores are calculated from the degraded classifier. We then evaluate the behavior of both the standard C2ST and the conformal C2ST across this interpolation path, providing a natural stress test of robustness.

**Experiment settings and baselines.** We benchmark the conformal C2ST against several well-established methods for assessing the quality of a neural posterior estimate. These include the C2ST (Lopez-Paz & Oquab, 2016), described in Section 2; SBC (Talts et al., 2018), which computes rank statistics for each marginal of $q(\theta \mid y)$ and checks for uniformity under the true posterior; TARP (Lemos et al., 2023), which uses randomly sampled reference points and distance-based statistics to construct a test that is both necessary and sufficient for posterior validity; and Discriminative Calibration (DC) which uses a multiclass classifier to get the strongest log-predictive-density (LPD) statistic (Yao & Domke, 2023, Algorithm 1) and AutoML (Kübler et al., 2023). We also include MMDFuse (Biggs et al., 2023), a kernel two-sample test that fuses multiple kernels without data splitting as a kernel discrepancy baseline. We exclude comparisons with $\ell$-C2ST (Linhart et al., 2023) since its primary objective is assessing the local performance of the NPE model at a specific conditioning input $x_o$, whereas our framework focuses on global testing across all $x$ with exact finite-sample guarantees.

**Evaluation setup.** We adapt each baseline to the validation task and compute rejection rates (at significance level

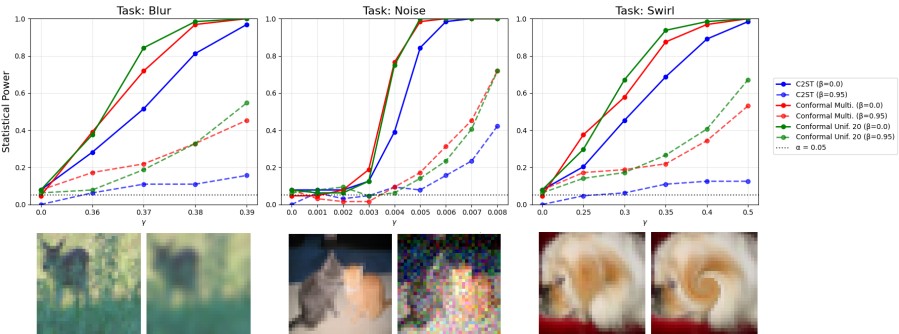

*Figure 3.* Conformal C2ST consistently outperforms standard C2ST across different tasks on CIFAR-10. Solid lines correspond to well-trained classifiers ($\beta = 0.0$), dashed lines to weaker classifiers ($\beta = 0.95$). Representative examples for each task are shown below.

$\alpha = 0.05$) across 200 independent trials. All methods share a common base of $N = 1000$ observations but differ in how many additional draws they require, and from which distribution (Table 1). SBC, TARP, and DC each need $m$ posterior samples from $q(\theta \mid y)$ per observation ($N \times m$ from $q$), the costly direction when sampling from $q$ is the bottleneck; DC is heavier still, training a multiclass classifier whose input dimension grows with $m$ and bootstrapping at evaluation. Conformal Uniform also pays an $m$-fold cost, but on the true joint $p(\theta, y)$—cheap in standard NPE, requiring only the simulator and prior—keeping its $q$-budget at $N$, while Conformal Multiple adds nothing, reusing a single shared calibration set to *exactly match* C2ST. MMDFuse, being kernel-based and split-free, simply takes the pooled train + test $2N$ samples from both $p$ and $q$.

*Table 1.* Sampling budgets per method (samples used to form the test statistic; $N = 1000$, $\alpha = 0.05$).

| Method | from $p$ | from $q$ |
|---|---|---|
| C2ST, AutoML, Conformal Multiple | $N$ | $N$ |
| Conformal Uniform($m$), $m \in \{1, 10, 50\}$ | $mN$ | $N$ |
| SBC ($m$=200), TARP ($m$=200) | $N$ | $mN$ |
| DC ($m$=10) | $N$ | $mN$ |
| MMDFuse | $2N$ | $2N$ |

All experiments use three random seeds, and we report the average results. We provide more details in Appendix B.2.

**Results.** As shown in Figure 2a, our conformal methods consistently achieve higher power than the standard C2ST and match or outperform SBC, DC and TARP across all types of model error. Most notably, DC fails to control Type I error, rendering it an invalid test in this setting. The practical benefit of our approach is its ability to detect subtle deviations from the true $p$. The Conformal Uniform and Conformal Multiple tests reliably identify misspecifications at low values of $\gamma$ where other methods fail. While most tests can spot large errors, our methods provide a much lower detection threshold, making them more useful for identifying the small but significant imperfections common in generative models.

In our second experiment, we tested each method's robustness to a weak classifier. For a fixed model error $\gamma$, we systematically degraded the classifier's performance from well-trained ($\beta = 0$) to random ($\beta = 1$), as described previously. Figure 2b reveals that the standard C2ST is quite brittle; its power collapses as soon as the classifier's quality degrades. In contrast, our conformal methods are highly robust, maintaining high power even when the classifier is far from optimal. This resilience stems from a fundamental advantage. While the C2ST requires an accurate classifier to draw a sharp decision boundary, our conformal methods only need the classifier's scores to provide a weak but informative ranking. This makes them far more reliable in practice, where perfectly trained classifiers are rarely available.

## 5.2 High-dimensional image experiments

To demonstrate the broad utility of the Conformal C2ST, we also evaluated it on the general-purpose task of detecting distributional shift in high-dimensional images. We used the CIFAR-10 dataset (Krizhevsky et al., 2009) to create a two-sample problem: distinguishing original, clean images from versions altered by one of three corruptions: Gaussian blur, swirl distortion, or additive Gaussian noise. A parameter $\gamma$ controlled the severity of the corruption. We compare the *clean* and *corrupted* distributions using C2ST, Conformal-multiple, and Conformal-uniform ($m$=20). Power is reported (i) as a function of corruption strength $\gamma$ with a well-trained classifier, and (ii) as a function of classifier quality $\beta$ at fixed $\gamma$ (larger $\beta$ denotes a weaker classifier). As summarized in Figure 3, the Conformal C2ST achieved the highest power across all corruption types and severity levels. Crucially, it maintained its superior performance even when the underlying classifier was weak, confirming its robustness and practical value for general-purpose generative model validation.

*Table 2.* Gravitational-lensing benchmark. Power (%) over 50 replicates. *(a)* Sensitivity at fixed classifier quality ($\beta = 0$) as the score is interpolated from exact ($\gamma = 0$) to fully biased ($\gamma = 1$). *(b)* Robustness at fixed bias ($\gamma = 1.0$) as the classifier is degraded toward random ($\beta \to 1$). Best valid result per row in **bold**; CM = Conformal Multiple, CU(50) = Conformal Uniform ($m = 50$). DC inflates Type-I error (see $\gamma = 0$) and is invalid here.

(a) Sensitivity: power (%) vs. bias $\gamma$ ($\beta = 0$).

| $\gamma$ | AutoML | C2ST | DC | MMDFuse | SBC | TARP | **CM** | **CU(50)** |
|---|---|---|---|---|---|---|---|---|
| 0.00 | 4 | 2 | 10 | 8 | 10 | 4 | 2 | 5 |
| 0.50 | 6 | 14 | 22 | 12 | 4 | 2 | **24** | 16 |
| 0.60 | 28 | 28 | 46 | 14 | 6 | 6 | 50 | **66** |
| 0.70 | 70 | 76 | 52 | 18 | 6 | 6 | 92 | **98** |
| 0.80 | 100 | 100 | 66 | 24 | 8 | 6 | 100 | 100 |
| 0.90 | 100 | 100 | 80 | 24 | 2 | 2 | 100 | 100 |
| 1.00 | 100 | 100 | 98 | 28 | 6 | 24 | 100 | 100 |

(b) Robustness: power (%) vs. degradation $\beta$ ($\gamma = 1.0$).

| $\beta$ | AutoML | C2ST | **CM** | **CU(50)** |
|---|---|---|---|---|
| 0.00 | 100 | 100 | 100 | 100 |
| 0.60 | 100 | 100 | 100 | 100 |
| 0.70 | 84 | 92 | 98 | **100** |
| 0.80 | 44 | 40 | 74 | **86** |
| 0.90 | 10 | 14 | **30** | 16 |
| 0.95 | 2 | 6 | **16** | 2 |

## 5.3 Gravitational lensing experiment

To test our methods on a real-data NPE problem, we evaluate on a scientific inverse problem, following the gravitational-lensing benchmark of Lemos et al. (2023). We place a Gaussian prior over $16 \times 16$ source images ($\theta \in \mathbb{R}^{256}$) drawn from galaxy data (PROBES), and use a linear lensing-style forward model $y = A\theta + \varepsilon$ comprising an affine warp, Gaussian blur, and additive Gaussian noise. Posterior samples are obtained from the reverse SDE of a score-based diffusion via conditional score decomposition.

Unlike Lemos et al. (2023), who compare only the exact and fully biased endpoints, we interpolate continuously between them in order to measure *how early* a bias is detected:

$$s_\gamma(\theta_t \mid x) = (1 - \gamma)\, s_{\text{true}}(\theta_t \mid x) + \gamma\, s_{\text{biased}}(\theta_t \mid x),$$

so that $\gamma = 0$ yields the exact posterior sampler and $\gamma = 1$ the fully biased one. Mirroring Section 5.1, we run two experiments: a *sensitivity* study that fixes a well-trained classifier ($\beta = 0$) and varies the bias $\gamma$, and a *robustness* study that fixes the fully biased sampler ($\gamma = 1.0$) and degrades the classifier toward random ($\beta \to 1$). In each replicate we use $n_{\text{true}} = n_{\text{fake}} = 2000$ with one posterior sample per $x$ for training, and report rejection rates over 50 independent test sets; MMDFuse uses the matched train+test budget.

**Results.** As shown in Table 2, the conformal tests stay calibrated near the nominal 5% level when the sampler is exact ($\gamma = 0$), whereas DC is inflated and hence invalid here. As the bias grows, both conformal variants detect it at smaller $\gamma$ than every baseline, while SBC and TARP remain near noise even at $\gamma = 1.0$, consistent with their known insensitivity to joint discrepancies. Under classifier degradation (panel b), ordinary C2ST and AutoML collapse quickly, while Conformal-Multiple and Conformal-Uniform(50) retain substantially more power. The trends from our controlled benchmarks thus carry over effectively to a real, high-dimensional scientific inverse problem.

## 6 Conclusion

We introduced the *Conformal C2ST*, which converts the scores of any classifier—however weak, biased, or overfit—into exact, finite-sample $p$-values. Validity follows from conformal exchangeability and holds for any scoring function, while power is governed by the classifier's ranking ability (its AUC) rather than its classification accuracy. We instantiated this idea in two variants—a uniform test with exact finite-sample validity, and a budget-matched multiple test—and proved that both retain non-trivial power under the alternative, degrading gracefully as $O(\varepsilon^{2/3})$ in the estimation error of the density ratio. Among classifier-based diagnostics, both variants—the exact uniform test and the budget-matched multiple test—attain state-of-the-art power, detecting subtle discrepancies that competitors miss while staying robust as the classifier degrades, across Gaussian perturbations, image corruptions, and a real gravitational-lensing problem. The broader message is: a near-Bayes-optimal classifier is not strictly required for reliable model validation—a weak but informative ranking suffices.

**Limitations.** Our work has several limitations. First our approach is designed to assess whether the learned posterior $q$ is a faithful approximation of the model's true posterior $p$. We do not address the separate, crucial problem of *statistical* model validation, where $p$ may not accurately reflect the real-world data-generating process in the $\mathcal{M}$-open scenario.

Our test is also *global*: it certifies whether $q$ matches $p$ over the full joint distribution but does not localize where a mismatch occurs. Moreover, our exact finite-sample guarantee (Lemma 3.1) covers only the uniform test; the budget-matched multiple test is only asymptotically valid. Relatedly, the Uniform variant's higher power relies on cheap draws from $p$ available in most NPE settings; for expensive or black-box simulators, however, these extra calibration samples are impractical, leaving only the budget-matched Multiple test. Finally, our theoretical analysis relies on bounding arguments that may be conservative; tighter techniques could potentially yield sharper error bounds.

## Impact Statement

This paper presents work whose goal is to advance the field of machine learning for scientific problems. There are many potential societal consequences of our work, none which we feel must be specifically highlighted here.

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

# A  Theoretical Results

In this section, we restate our key lemmas in full and provide rigorous proofs to support our theoretical claims.

## A.1  Proofs of Lemma 3.2 and 3.3

**Lemma 3.2** (Full statement). *Let $p$ and $q$ be two probability densities on a measurable space $\mathcal{X}$, such that $p(x) > 0$ and $q(x) > 0$ for all $x \in \mathcal{X}$. Let $r(x) := \frac{p(x)}{q(x)}$ denote the oracle density ratio. For any measurable scoring function $s : \mathcal{X} \to \mathbb{R}$, define the area under the ROC curve (AUC) as*

$$\mathrm{AUC}(s) := \mathbb{P}[s(X) > s(\tilde{X})] + \frac{1}{2}\mathbb{P}[s(X) = s(\tilde{X})],$$

*where $X \sim p$ and $\tilde{X} \sim q$ are independent.*

*Then $\mathrm{AUC}(s) \leq \mathrm{AUC}(r)$, with equality if and only if there exists a strictly increasing function $h$ such that $s(x) = h(r(x))$ for $q$-almost every $x$.*

*Proof.* Define

$$\phi_s(x, \tilde{x}) := \mathbb{I}[s(x) > s(\tilde{x})] + \frac{1}{2}\mathbb{I}[s(x) = s(\tilde{x})],$$

so that

$$\mathrm{AUC}(s) = \iint \phi_s(x, \tilde{x})\, p(x)q(\tilde{x})\, dx d\tilde{x}.$$

Define the antisymmetric part

$$a_s(x, \tilde{x}) := \phi_s(x, \tilde{x}) - \phi_s(\tilde{x}, x) \in \{-1, 0, 1\}.$$

Note that $\phi_s(x, \tilde{x}) + \phi_s(\tilde{x}, x) = 1$, hence

$$\phi_s(x, \tilde{x}) = \frac{1}{2} + \frac{1}{2}a_s(x, \tilde{x}),$$

and therefore

$$\mathrm{AUC}(s) = \frac{1}{2} + \frac{1}{2}\iint a_s(x, \tilde{x})\, p(x)q(\tilde{x})\, dx d\tilde{x}.$$

Define the antisymmetric function

$$W(x, \tilde{x}) := p(x)q(\tilde{x}) - p(\tilde{x})q(x),$$

so that

$$\iint a_s(x, \tilde{x})\, p(x)q(\tilde{x})\, dx d\tilde{x} = \frac{1}{2}\iint a_s(x, \tilde{x})W(x, \tilde{x})\, dx d\tilde{x},$$

and thus

$$\mathrm{AUC}(s) = \frac{1}{2} + \frac{1}{4}\iint a_s(x, \tilde{x})W(x, \tilde{x})\, dx d\tilde{x}.$$

Now for each $(x, \tilde{x})$, the value $a_s(x, \tilde{x}) \in \{-1, 0, 1\}$ that maximizes the product $a_s(x, \tilde{x})W(x, \tilde{x})$ is $\mathrm{sign}(W(x, \tilde{x}))$. Therefore, the function

$$a^*(x, \tilde{x}) := \mathrm{sign}(W(x, \tilde{x}))$$

maximizes the integral.

Next, define the likelihood ratio $r(x) := \frac{p(x)}{q(x)}$. Then

$$r(x) > r(\tilde{x}) \iff \frac{p(x)}{q(x)} > \frac{p(\tilde{x})}{q(\tilde{x})} \iff p(x)q(\tilde{x}) > p(\tilde{x})q(x) \iff W(x, \tilde{x}) > 0,$$

so

$$a_r(x, \tilde{x}) := \mathrm{sign}(r(x) - r(\tilde{x})) = \mathrm{sign}(W(x, \tilde{x})) = a^*(x, \tilde{x}).$$

Thus, $a_r$ maximizes the integral, and

$$\text{AUC}(s) \leq \frac{1}{2} + \frac{1}{4} \iint |W(x, \tilde{x})| \, dx d\tilde{x} = \text{AUC}(r).$$

Finally, equality occurs if and only if $a_s(x, \tilde{x}) = a_r(x, \tilde{x})$ for almost all $(x, \tilde{x})$, which implies that $s(x) > s(\tilde{x}) \iff r(x) > r(\tilde{x})$, i.e., $s = h(r)$ for some strictly increasing function $h$ almost everywhere. $\qquad\square$

**Corollary A.1** (Lower Bound on AUC via Total Variation). *Under the setup of the previous lemma, let* $\text{TV}(p, q) := \frac{1}{2} \int |p(x) - q(x)| dx$ *denote the total variation distance between $p$ and $q$. Then*

$$\text{AUC}(r) \geq \frac{1 + \text{TV}(p, q)}{2}.$$

*Proof.* Consider the binary Bayes classifier

$$\phi^*(x) := \mathbb{I}[r(x) > 1] = \mathbb{I}\left[\frac{p(x)}{q(x)} > 1\right],$$

which is known to be the most powerful test at level $\alpha = \frac{1}{2}$. Its classification accuracy is:

$$\mathbb{P}[\phi^*(X) = 1] \cdot \frac{1}{2} + \mathbb{P}[\phi^*(\tilde{X}) = 0] \cdot \frac{1}{2} = \frac{1}{2} + \frac{1}{2}\text{TV}(p, q).$$

Now note that if we treat $\phi^* \in \{0, 1\}$ as a scoring function, the AUC of this classifier is:

$$\text{AUC}(\phi^*) = \mathbb{P}[\phi^*(X) > \phi^*(\tilde{X})] + \frac{1}{2}\mathbb{P}[\phi^*(X) = \phi^*(\tilde{X})] \leq \text{AUC}(r),$$

since $\phi^*$ is a thresholding of $r$, and AUC is maximized by ranking with $r$.

But:

$$\text{AUC}(\phi^*) = \frac{1}{2} + \frac{1}{2}\text{TV}(p, q),$$

so we conclude:

$$\text{AUC}(r) \geq \frac{1}{2} + \frac{1}{2}\text{TV}(p, q) = \frac{1 + \text{TV}(p, q)}{2}.$$

$\qquad\square$

**Lemma 3.3** (Expected conformal p-value under the alternative). *Let $p$ and $q$ be as defined above, and let $U$ denote the conformal p-value computed using the oracle density ratio $r(x) = p(x)/q(x)$ as the score function, as defined in (1), with $m$ calibration samples drawn from $p$ for each test point drawn from $q$. Then, under the alternative hypothesis $H_1 : p \neq q$, we have*

$$\mathbb{E}[U] = 1 - \text{AUC}(r) \leq \frac{1}{2} - \frac{1}{2}\text{TV}(p, q) < \frac{1}{2},$$

*in the limit as $m \to \infty$.*

*Proof.* We formalize our discussion in Section 3.1. First, by the strong law of large numbers, we have

$$U \xrightarrow[m \to \infty]{\text{a.s.}} \mathbb{P}_{X \sim p}\left[r(X) < r(\tilde{X})\right] + \xi \mathbb{P}_{X \sim p}\left[r(X) = r(\tilde{X})\right], \quad \text{where } \xi \sim \text{Unif}(0, 1).$$

Next, we take expectation with respect to $\tilde{X} \sim q$ and $\xi \sim \text{Unif}(0, 1)$. However, since $U \leq 1$, Dominated Convergence Theorem gives that

$$\mathbb{E}[U] \xrightarrow[m \to \infty]{\text{a.s.}} \mathbb{P}\left[r(X) < r(\tilde{X})\right] + \frac{1}{2}\mathbb{P}\left[r(X) = r(\tilde{X})\right]$$
$$= 1 - \text{AUC}(r)$$

The result follows immediately from Corollary A.1, since $0 < \text{TV}(p, q) \leq 1$ under $H_1 : p \neq q$. $\qquad\square$

We note that (Hu & Lei, 2024) present a related result from an information-theoretic perspective, showing that the variability of the density ratio $r(x)$ under $q$ controls the deviation of conformal p-values from uniformity. In contrast, our focus is on quantifying the relationship between a classifier's discriminative ability and the statistical power of the resulting two-sample test. While the underlying intuition is similar, our formulation offers a more direct and operational perspective, grounded in the ranking statistic captured by the AUC score. For completeness, we restate their relevant lemma below.

**Lemma A.2** (Restated from (Hu & Lei, 2024)). *Under the alternative $H_1 : p \neq q$, we have*

$$\mathbb{E}[U] = \frac{1}{2} - \frac{1}{4}\mathbb{E}_{X,X'\sim q}\left[|r(X) - r(X')|\right] < \frac{1}{2} \quad as\ m \to \infty,$$

*where $X, X'$ are i.i.d. draws from $q$.*

## A.2   Proof of Theorem 3.5

*Proof.* We start by taking the expectation of our conformal p-value wrt the tie breaking uniform variable $\xi$:

$$2\,\mathbb{E}_\xi[\hat{U}] = \frac{1}{m+1}\left(\sum_{i=1}^m \mathbb{I}(\hat{r}(X_i) < \hat{r}(\tilde{X})) + \sum_{i=1}^m \mathbb{I}(\hat{r}(X_i) \leq \hat{r}(\tilde{X})) + 1\right).$$

By the Strong Law of Large Numbers (SLLN), as $m \to \infty$, we have:

$$2\,\mathbb{E}_\xi[\hat{U}] \to \mathbb{E}[\mathbb{I}(\hat{r}(X) < \hat{r}(\tilde{X})) \mid \tilde{X}] + \mathbb{E}[\mathbb{I}(\hat{r}(X) \leq \hat{r}(\tilde{X})) \mid \tilde{X}].$$

Taking expectation over $\tilde{X}$ and applying the Dominated Convergence Theorem (since $\hat{U} \leq 1$):

$$2\,\mathbb{E}[\hat{U}] \to \mathbb{E}[\mathbb{I}(\hat{r}(X) < \hat{r}(\tilde{X}))] + \mathbb{E}[\mathbb{I}(\hat{r}(X) \leq \hat{r}(\tilde{X}))].$$

Now define $\tilde{X}, \tilde{X}' \overset{\text{iid}}{\sim} q$. Using importance reweighting, we write:

$$2\,\mathbb{E}[\hat{U}] = \mathbb{E}\left[r(\tilde{X}') \cdot \mathbb{I}(\hat{r}(\tilde{X}') < \hat{r}(\tilde{X}))\right] + \mathbb{E}\left[r(\tilde{X}') \cdot \mathbb{I}(\hat{r}(\tilde{X}') \leq \hat{r}(\tilde{X}))\right].$$

Since $\mathbb{E}[r(\tilde{X}')] = 1$, this becomes:

$$2\,\mathbb{E}[\hat{U}] = 1 - \mathbb{E}[(r(\tilde{X}') - r(\tilde{X})) \cdot \mathbb{I}(\hat{r}(\tilde{X}') > \hat{r}(\tilde{X}))].$$

Define:

$$Z := r(\tilde{X}') - r(\tilde{X}), \quad \hat{Z} := \hat{r}(\tilde{X}') - \hat{r}(\tilde{X}), \quad \Delta := \hat{Z} - Z.$$

Then:

$$\mathbb{E}[\hat{U}] = \mathbb{E}[U] + \frac{1}{2}\delta, \quad \text{where} \quad \delta := \frac{1}{2}\mathbb{E}[|Z|] - \mathbb{E}[Z \cdot \mathbb{I}(\hat{Z} > 0)].$$

By symmetry of $Z$,

$$\delta = \frac{1}{2}\mathbb{E}[|Z|] - \mathbb{E}[Z \cdot \mathbb{I}(\hat{Z} > 0)] = \mathbb{E}[Z(\mathbb{I}(Z > 0) - \mathbb{I}(\hat{Z} > 0))].$$

Define events:

$$A := \{Z > 0, \hat{Z} \leq 0\}, \quad B := \{Z \leq 0, \hat{Z} > 0\}.$$

Then:

$$\delta = \mathbb{E}[Z \cdot \mathbb{I}_A] - \mathbb{E}[Z \cdot \mathbb{I}_B] \leq \mathbb{E}[|Z| \cdot \mathbb{I}_{|\Delta| \geq |Z|}].$$

For any threshold $t > 0$, we decompose:

$$\delta \leq \mathbb{E}[|Z| \cdot \mathbb{I}(|Z| \leq t)] + \mathbb{E}[|Z| \cdot \mathbb{I}(|\Delta| > t)].$$

The second term is bounded by Cauchy–Schwarz and Markov:

$$\mathbb{E}[|Z| \cdot \mathbb{I}(|\Delta| > t)] \leq \sqrt{\mathbb{E}[Z^2]} \cdot \sqrt{\mathbb{P}(|\Delta| > t)} \leq \sqrt{\mathbb{E}[Z^2]} \cdot \frac{2\varepsilon}{t}.$$

Assuming the density $f_Z$ of $Z$ is bounded near zero by $C$, we have:

$$\mathbb{E}[|Z| \cdot \mathbb{I}(|Z| \leq t)] \leq 2C \int_0^t z \, dz = Ct^2.$$

Combining,

$$\delta \leq Ct^2 + \frac{2\sqrt{\mathbb{E}[Z^2]}\,\varepsilon}{t}.$$

Minimizing the RHS by choosing $t = \left( \frac{2\sqrt{\mathbb{E}[Z^2]}\,\varepsilon}{C} \right)^{1/3}$, we obtain:

$$\delta = O(\varepsilon^{2/3}).$$

$\square$

## A.3   Variance scaling of conformal p-value

In this section we assume that the score function $s$ gives continuous outputs, so that the probability of ties is zero. Note that the assumption is ubiquitously satisfied in practical settings when classifiers are trained using neural networks.

**Lemma A.3** (Variance of Conformal p-value). *Let $\hat{U}$ be the conformal p-value for a test point $\tilde{X} \sim q$. The variance of $\hat{U}$, conditioned on the test point, is:*

$$Var(\hat{U}|\tilde{X}) = \frac{mU(1-U)}{(m+1)^2} = \mathcal{O}(1/m)$$

*where $U = P(s(X) \leq s(\tilde{X})|X \sim p)$ is the true, population-level p-value (assuming no ties).*

*Proof.* By definition, the conformal p-value (assuming no ties for simplicity) is given by:

$$\hat{U} = \frac{1}{m+1}\left( 1 + \sum_{i=1}^m \mathbb{I}\{s(X_i) \leq s(\tilde{X})\} \right)$$

where $\{X_i\}_{i=1}^m$ are i.i.d. calibration samples from the distribution $p$.

When we condition on the test point $\tilde{X}$, the score $s(\tilde{X})$ becomes a fixed value. Let's define a set of indicator random variables $B_i$ for $i = 1, \ldots, m$:

$$B_i = \mathbb{I}\{s(X_i) \leq s(\tilde{X})\}$$

Since the calibration samples $X_i$ are drawn i.i.d. from $p$, the variables $B_i$ are i.i.d. Bernoulli random variables. The probability of success for each $B_i$ is:

$$P(B_i = 1) = P(s(X_i) \leq s(\tilde{X})) = U$$

where $U$ is the population-level p-value as defined in the lemma statement. Thus, $B_i \sim \text{Bernoulli}(U)$.

We can now express $\hat{U}$ in terms of these Bernoulli variables:

$$\hat{U} = \frac{1}{m+1}\left( 1 + \sum_{i=1}^m B_i \right)$$

The variance of $\hat{U}$ conditioned on $\tilde{X}$ is:

$$\begin{aligned}
Var(\hat{U}|\tilde{X}) &= Var\left( \frac{1}{m+1}\left( 1 + \sum_{i=1}^m B_i \right) \right)\\
&= \left( \frac{1}{m+1} \right)^2 Var\left( 1 + \sum_{i=1}^m B_i \right)\\
&= \frac{1}{(m+1)^2} Var\left( \sum_{i=1}^m B_i \right)
\end{aligned}$$

Since the $B_i$ are i.i.d., the variance of their sum is the sum of their variances:

$$\text{Var}\left(\sum_{i=1}^{m} B_i\right) = \sum_{i=1}^{m} \text{Var}(B_i)$$

The variance of a Bernoulli($U$) random variable is $U(1-U)$. Therefore:

$$\sum_{i=1}^{m} \text{Var}(B_i) = \sum_{i=1}^{m} U(1-U) = mU(1-U)$$

Substituting this back, we get the final result:

$$\text{Var}(\hat{U}|\tilde{X}) = \frac{mU(1-U)}{(m+1)^2}$$

As the number of calibration samples $m \to \infty$, the variance behaves as:

$$\frac{mU(1-U)}{(m+1)^2} \approx \frac{mU(1-U)}{m^2} = \frac{U(1-U)}{m} = \mathcal{O}(1/m)$$

This completes the proof. $\qquad\square$

Hence, the resulting $\mathcal{O}(1/m)$ scaling of the variance of conformal p-value ensures that the test power quickly rises as the p-values converge quickly to their true non-uniform values under the alternative when one increases the number of calibration points $m$.

### A.4 Multiple test analysis

In this section, we summarize the multiple conformal testing procedure proposed by (Hu & Lei, 2024), which accounts for the dependence among p-values arising from the use of a shared calibration set. Let $X_1, \ldots, X_{n_p} \overset{\text{i.i.d.}}{\sim} p$ (calibration) and $\tilde{X}_1, \ldots, \tilde{X}_{n_q} \overset{\text{i.i.d.}}{\sim} q$ (test), independent. Let $r(x) = p(x)/q(x)$. Under the NPE setting, where the marginal distribution of $y$ is assumed to be the same under both the true and approximate joint distributions (or under the unconditional setting), we derive a simplified form of their test statistic:

$$\hat{T} = \frac{\frac{1}{2} - \frac{1}{n_q}\sum_{j=1}^{n_q} \hat{U}_j}{\hat{\sigma}/\sqrt{n_p}}$$

where $\hat{U}_j := \frac{1}{n_p}\left(\sum_{i=1}^{n_p} \mathbb{I}\left\{\hat{r}(X_i) < \hat{r}(\tilde{X}_j)\right\} + \xi_j \cdot \sum_{i=1}^{n_p} \mathbb{I}\left\{\hat{r}(X_i) = \hat{r}(\tilde{X}_j)\right\}\right)$ is obtained from the entire calibration set (common to all test points) with $\hat{r}(\cdot)$ as the scoring function, and $\hat{\sigma}$ is the asymptotic estimated standard deviation of $\frac{\sqrt{n_p}}{n_q}\sum_{j=1}^{n_q} \hat{U}_j$. We also adapt their expression for the variance estimate to the same-marginal setting. Let $\hat{F}$ be the empirical CDF of $\left\{\hat{r}(\tilde{X}_j) : j \in [n_q]\right\}$ and $\hat{F}_-$ be its left limit. Then the variance estimate is given by:

$$\hat{\sigma}^2 = \hat{\sigma}_1^2 + \frac{n_p}{12 \cdot n_q}$$

where $\hat{\sigma}_1^2$ is the empirical variance of $\left\{\hat{F}_{1/2}(\hat{r}(X_i)) : i \in [n_p]\right\}$ and $\hat{F}_{1/2}(t) = \left(\hat{F}(t) + \hat{F}_-(t)\right)/2$.

#### A.4.1 FRAMING AS A U-STATISTIC AND CLT

Let $s : \mathcal{X} \to \mathbb{R}$ be a *fixed* score (either $r$ or $\hat{r}$) independent of the evaluation samples.

**Average of $p$-values is a U-statistic**   Let $\{\xi_j\}_{1 \le j \le n_q} \overset{\text{i.i.d.}}{\sim} \text{Unif}(0,1)$, independent of all data. Define the randomized kernel and its mid-rank expectation:

$$h_s(x, y; \xi) := \mathbb{I}\{s(x) < s(y)\} + \xi\,\mathbb{I}\{s(x) = s(y)\},$$
$$\bar{h}_s(x, y) := \mathbb{E}_\xi\big[h_s(x, y; \xi)\big] = \mathbb{I}\{s(x) < s(y)\} + \tfrac{1}{2}\mathbb{I}\{s(x) = s(y)\}.$$

For a test point $\tilde{X}_j$, its conformal $p$-value (with shared calibration and per-pair tie-breaking) is

$$\hat{U}_j(s) = \frac{1}{n_p} \sum_{i=1}^{n_p} h_s(X_i, \tilde{X}_j; \xi_j).$$

The average is

$$\bar{U}_s^{(\xi)} = \frac{1}{n_q} \sum_{j=1}^{n_q} \hat{U}_j(s) = \frac{1}{n_p n_q} \sum_{i=1}^{n_p} \sum_{j=1}^{n_q} h_s(X_i, \tilde{X}_j; \xi_j).$$

Define the mid-rank average $\bar{U}_s := (n_p n_q)^{-1} \sum_{i,j} \bar{h}_s(X_i, \tilde{X}_j)$.

Let $\psi_s := \mathbb{E}[\bar{h}_s(X, \tilde{X})] = 1 - \text{AUC}(s)$ and $\sigma_s^2$ be the asymptotic variance in the CLT for $\sqrt{n_p}(\bar{U}_s - \psi_s)$ (see Proposition A.5). The multiple statistic is

$$\hat{T}(s) := \frac{\tfrac{1}{2} - \bar{U}_s^{(\xi)}}{\hat{\sigma}(s)/\sqrt{n_p}},$$

with $\hat{\sigma}(s) \overset{p}{\to} \sigma_s$.

First, we use the standard Hoeffding decomposition to derive a CLT for the mid-rank average U-statistic $\bar{U}_s$.

**Lemma A.4** (Hoeffding decomposition). *Define, for fixed $s$,*

$$\phi_{p,s}(x) := \mathbb{E}[\bar{h}_s(x, \tilde{X})] - \psi_s,$$
$$\phi_{q,s}(y) := \mathbb{E}[\bar{h}_s(X, y)] - \psi_s,$$
$$\phi_{0,s}(x, y) := \bar{h}_s(x, y) - \psi_s - \phi_{p,s}(x) - \phi_{q,s}(y).$$

*Then,*

$$\bar{U}_s - \psi_s = \frac{1}{n_p} \sum_{i=1}^{n_p} \phi_{p,s}(X_i) + \frac{1}{n_q} \sum_{j=1}^{n_q} \phi_{q,s}(\tilde{X}_j) + R_{n_p,n_q}(s), \tag{4}$$

$$\mathbb{E}[\phi_{p,s}(X)] = \mathbb{E}[\phi_{q,s}(\tilde{X})] = \mathbb{E}[\phi_{0,s}(X, \tilde{X})] = 0,$$
$$\mathbb{E}[\phi_{0,s}(X, \tilde{X}) \mid X] = \mathbb{E}[\phi_{0,s}(X, \tilde{X}) \mid \tilde{X}] = 0. \tag{5}$$

*Moreover,*

$$R_{n_p,n_q}(s) := \frac{1}{n_p n_q} \sum_{i=1}^{n_p} \sum_{j=1}^{n_q} \phi_{0,s}(X_i, \tilde{X}_j)$$

$$Var(R_{n_p,n_q}(s)) \le \frac{C_0}{n_p n_q} \qquad \text{for a universal constant } C_0 < \infty. \tag{6}$$

*Proof.* Equation (4) is the standard two-sample Hoeffding decomposition obtained by adding and subtracting $\mathbb{E}[\bar{h}_s(X, \tilde{X}) \mid X]$ and $\mathbb{E}[\bar{h}_s(X, \tilde{X}) \mid \tilde{X}]$ termwise and regrouping. The centering identities in (5) follow directly from the definitions. For (6), boundedness $0 \le \bar{h}_s \le 1$ implies $|\phi_{0,s}| \le 2$, and standard variance bounds for degenerate two-sample U-statistics (or a direct Efron–Stein/Poincaré argument) yield $\text{Var}(R_{n_p,n_q}(s)) \le C_0/(n_p n_q)$. $\qquad\square$

**Proposition A.5** (CLT with explicit variance). *Let $\sigma_{p,s}^2 := Var(\phi_{p,s}(X))$, $\sigma_{q,s}^2 := Var(\phi_{q,s}(\tilde{X}))$. Then*

$$\sqrt{n_p}(\bar{U}_s - \psi_s) \Rightarrow N\big(0,\, \sigma_s^2\big), \qquad \sigma_s^2 = \sigma_{p,s}^2 + \frac{n_p}{n_q} \sigma_{q,s}^2.$$

*Moreover, $Var(R_{n_p,n_q}(s)) \le C/(n_p n_q)$ for a universal $C$, so $\sqrt{n_p} R_{n_p,n_q}(s) \to 0$ in $L_2$.*

*Proof.* Multiply (4) by $\sqrt{n_p}$, yielding

$$\sqrt{n_p}(\bar{U}_s - \psi_s) = \frac{1}{\sqrt{n_p}} \sum_{i=1}^{n_p} \phi_{p,s}(X_i) + \sqrt{\frac{n_p}{n_q}} \cdot \frac{1}{\sqrt{n_q}} \sum_{j=1}^{n_q} \phi_{q,s}(\tilde{X}_j) + \sqrt{n_p}\, R_{n_p, n_q}(s).$$

By the Lindeberg CLT for i.i.d. sums, the two normalized sums converge jointly to independent Gaussians with variances $\sigma_{p,s}^2$ and $(n_p/n_q)\sigma_{q,s}^2$. Independence across the $X$- and $\tilde{X}$-blocks gives variance additivity. The degenerate remainder satisfies $\mathbb{E}[\left(\sqrt{n_p} R_{n_p, n_q}(s)\right)^2] \leq C_0\, n_p/(n_p n_q) = C_0/n_q \to 0$ by (6). $\quad\square$

Now, under Assumption 4.1, we argue that one can essentially ignore the "tie-noise".

**Proposition A.6** (Tie-noise: decomposition, variance, and scale). *Let*

$$M_{n_p, n_q}(s) := \frac{1}{n_p n_q} \sum_{j=1}^{n_q} \left(\xi_j - \tfrac{1}{2}\right) K_j, \qquad N_{\text{tie}}(s) := \sum_{i=1}^{n_p} \sum_{j=1}^{n_q} \mathbb{I}\{s(X_i) = s(\tilde{X}_j)\},$$

*where $K_j := \sum_{i=1}^{n_p} \mathbb{I}\{s(X_i) = s(\tilde{X}_j)\}$ and $\{\xi_j\} \overset{i.i.d.}{\sim} \text{Unif}(0,1)$ are independent of all data. Denote*

$$\bar{U}_s^{(\xi)} := \frac{1}{n_p n_q} \sum_{i=1}^{n_p} \sum_{j=1}^{n_q} \left(\mathbb{I}\{s(X_i) < s(\tilde{X}_j)\} + \xi_{ij}\, \mathbb{I}\{s(X_i) = s(\tilde{X}_j)\}\right), \quad \bar{U}_s := \mathbb{E}_\xi\left[\bar{U}_s^{(\xi)}\right].$$

*Then:*

(i) *Decomposition and centering.* $\bar{U}_s^{(\xi)} = \bar{U}_s + M_{n_p, n_q}(s)$ *and* $\mathbb{E}\left[M_{n_p, n_q}(s) \mid X, \tilde{X}\right] = 0.$

(ii) *Scale under Assumption 4.1. If ties occur at rate $O_p(1)$, then*

$$\sqrt{n_p}\, M_{n_p, n_q}(s) = O_p\left(\frac{1}{n_q \sqrt{n_p}}\right).$$

*Proof.* (i) Linearity in $\xi_j$ gives

$$\bar{U}_s^{(\xi)} - \bar{U}_s = \frac{1}{n_p n_q} \sum_{i,j} \left(\xi_j - \tfrac{1}{2}\right) \mathbb{I}\{s(X_i) = s(\tilde{X}_j)\} = M_{n_p, n_q}(s),$$

hence the decomposition; conditional mean zero follows since $\mathbb{E}[\xi_j - 1/2] = 0$.

(ii) Conditional on $(X, \tilde{X})$, the summands are independent with $\text{Var}(\xi_j - \tfrac{1}{2}) = 1/12$ and vanish off ties, so

$$\mathbb{E}\left[M_{n_p, n_q}(s)^2 \mid X, \tilde{X}\right] = \text{Var}\left(M_{n_p, n_q}(s) \mid X, \tilde{X}\right) = \frac{1}{(n_p n_q)^2} \sum_{j=1}^{n_q} \frac{1}{12} K_j^2 \leq \frac{1}{12} \cdot \frac{N_{\text{tie}}^2(s)}{(n_p n_q)^2}.$$

Taking expectations and using $N_{\text{tie}}(s) = O_p(1)$ (under Assumption 4.1), Chebyshev's inequality yields the desired result. $\quad\square$

The following lemma now decomposes our multiple test statistic into three terms which we analyze later:

**Lemma A.7** (Expansion of $\hat{T}(s)$). *For any fixed $s$,*

$$\hat{T}(s) = \frac{\sqrt{n_p}}{\sigma_s}\left(\tfrac{1}{2} - \psi_s\right) - \frac{\sqrt{n_p}}{\sigma_s}(\bar{U}_s - \psi_s) - \underbrace{\frac{\sqrt{n_p}}{\hat{\sigma}(s)} M_{n_p, n_q}(s)}_{\text{tie-noise}} + O_p(1), \tag{7}$$

*where $\frac{\sqrt{n_p}}{\sigma_s}(\bar{U}_s - \psi_s) \Rightarrow \mathcal{N}(0,1)$ and the tie-noise term is $O_p(n_q^{-1} n_p^{-1/2})$.*

*Proof.* By construction, $\bar{U}_s^{(\xi)} = \bar{U}_s + M_{n_p, n_q}(s)$. Now write

$$\hat{T}(s) = \frac{\frac{1}{2} - \bar{U}_s^{(\xi)}}{\hat{\sigma}(s)/\sqrt{n_p}} = \frac{\sqrt{n_p}}{\hat{\sigma}(s)}\left(\frac{1}{2} - \psi_s\right) - \frac{\sqrt{n_p}}{\hat{\sigma}(s)}\left(\bar{U}_s - \psi_s\right) - \frac{\sqrt{n_p}}{\hat{\sigma}(s)} M_{n_p, n_q}(s).$$

Replace $\hat{\sigma}(s)$ by $\sigma_s$ using $\hat{\sigma}(s) \xrightarrow{p} \sigma_s$ and Slutsky twice, and apply Proposition A.5 to the mid-rank fluctuation and Proposition A.6 to tie-noise. Replacing minus signs by absorbing into terms yields (7). $\square$

### A.4.2 VALIDITY UNDER THE NULL

Throughout this section we work under Assumptions 3.4, 4.1, and the notation introduced earlier.

We first show that for any choice of the score function $s$ satisfying our assumptions, the test is asymptotically valid, i.e., $\hat{T}(s)$ has a normal distribution under the null.

**Theorem A.8** (Null normality). *Under $H_0 : p = q$,*

$$\hat{T}(s) = \frac{\sqrt{n_p}\left(\frac{1}{2} - \bar{U}_s\right)}{\sigma_s} + O_p\left(\frac{1}{\sqrt{n_q}}\right) + o_p(1). \tag{8}$$

*In particular, $\hat{T}(s) \Rightarrow N(0, 1)$.*

*Proof.* Under $H_0 : p = q$, $\psi_s = 1 - \text{AUC}(s) = 0.5$ by definition. Substitute in the expansion of (7) and use Proposition A.6 to obtain $\sqrt{n_p}\, M_{n_p, n_q}(s) = O_p\left(\frac{1}{n_q \sqrt{n_p}}\right)$. Divide by $\hat{\sigma}(s)$ and use $\hat{\sigma}(s) \to \sigma_s$; combine with Proposition A.5. $\square$

### A.4.3 PROOF OF THEOREM 4.2

*Proof.* We analyze the asymptotic behavior of the test statistic by decomposing it into a deterministic drift term, a stochastic fluctuation term, and a tie-breaking noise term. Recall the definition of the statistic for a generic score $s$ defined in (4):

$$\hat{T}(s) = \frac{1/2 - \overline{U}_s^{(\xi)}}{\hat{\sigma}(s)/\sqrt{n_p}}$$

where $\overline{U}_s^{(\xi)}$ is the randomized U-statistic. We decompose the numerator around its population mean $\psi_s = \mathbb{E}[\overline{U}_s] = 1 - \text{AUC}(s)$. Let $\overline{U}_s$ denote the mid-rank U-statistic (without tie-breaking noise). We write:

$$\hat{T}(s) = \frac{\sqrt{n_p}}{\hat{\sigma}(s)}\left(\frac{1}{2} - \psi_s\right) - \frac{\sqrt{n_p}}{\hat{\sigma}(s)}(\overline{U}_s - \psi_s) - \frac{\sqrt{n_p}}{\hat{\sigma}(s)}(\overline{U}_s^{(\xi)} - \overline{U}_s)$$

Using the consistency of the variance estimator ($\hat{\sigma}(s) \xrightarrow{p} \sigma_s$) and the definitions from the U-statistic CLT (Proposition A.5) and tie-noise bound (Proposition A.6), this simplifies to:

$$\hat{T}(s) = \frac{\sqrt{n_p}}{\sigma_s}\Delta_s - Z_s - \mathcal{O}_p(n_q^{-1/2}) + o_p(1)$$

where $\Delta_s = \text{AUC}(s) - 1/2$ is the drift, and $Z_s = \frac{\sqrt{n_p}}{\sigma_s}(\overline{U}_s - \psi_s)$ is the standardized fluctuation which converges weakly to $\mathcal{N}(0, 1)$. Without loss of generality, assume $n_p \gtrsim n_q$ (the opposite case is symmetric):

**Scaling of the Oracle Statistic.** For the oracle score $r$, under the alternative, $\Delta_r > 0$. From Proposition A.5, the variance scales as $\sigma_r^2 = \sigma_{p,r}^2 + \frac{n_p}{n_q}\sigma_{q,r}^2$. In the limit $n_p, n_q \to \infty$, we have $\frac{\sqrt{n_p}}{\sigma_r} \sim \frac{\sqrt{n_q}}{\sigma_{q,r}}$. Therefore:

$$\hat{T}(r) = \frac{\sqrt{n_q}}{\sigma_{q,r}}\Delta_r + \mathcal{O}_p(1) = \mathcal{O}_p(\sqrt{n_q}\Delta_r)$$

**Analysis of the Oracle-Estimate Gap.** We compare the estimated statistic $\hat{T}(\hat{r})$ to the oracle $\hat{T}(r)$ by examining the difference:

$$\hat{T}(\hat{r}) - \hat{T}(r) = \underbrace{\left[ \frac{\sqrt{n_p}}{\sigma_{\hat{r}}} \Delta_{\hat{r}} - \frac{\sqrt{n_p}}{\sigma_r} \Delta_r \right]}_{\text{Drift Gap}} - \underbrace{[Z_{\hat{r}} - Z_r]}_{\text{Fluctuation Gap}} + o_p(1)$$

1. Drift Gap: By Theorem 3.5, the ranking error is bounded by the estimation error: $|\Delta_{\hat{r}} - \Delta_r| = |\text{AUC}(\hat{r}) - \text{AUC}(r)| \leq C\varepsilon^{2/3}$. Combined with the variance scaling, the drift gap behaves as $\mathcal{O}_p(\sqrt{n_q}\varepsilon^{2/3})$.

2. Fluctuation Gap: The term $Z_{\hat{r}} - Z_r$ represents the difference between two centered U-statistics scaled to unit variance. The kernel of the U-statistic, $h(x,y) = \mathbb{I}(s(x) < s(y)) + \frac{1}{2}\mathbb{I}(s(x) = s(y))$, is bounded in $[0,1]$. Consequently, the difference kernel $D(x,y) = h_r(x,y) - h_{\hat{r}}(x,y)$ is bounded by 1. The variance of the difference is bounded by the second moment of the kernel difference, which is at most $\mathcal{O}(1)$. Thus, $Z_{\hat{r}} - Z_r = \mathcal{O}_p(1)$.

**Relative Error.** Combining these components, the relative error is:

$$\frac{\hat{T}(\hat{r}) - \hat{T}(r)}{\hat{T}(r)} = \frac{\mathcal{O}_p(\sqrt{n_q}\varepsilon^{2/3}) + \mathcal{O}_p(1)}{\mathcal{O}_p(\sqrt{n_q}\Delta_r)}$$

In the asymptotic limit as $n_q \to \infty$, the fluctuation term $\mathcal{O}_p(n_q^{-1/2})$ vanishes. Assuming non-trivial power ($\Delta_r > 0$), the relative error is dominated by the estimation quality:

$$\frac{\hat{T}(\hat{r}) - \hat{T}(r)}{\hat{T}(r)} = \mathcal{O}_p(\varepsilon^{2/3})$$

This completes the proof. $\square$

# B  Experiment Details

## B.1  Benchmarking with perturbed Gaussians

Below, we include perturbations from (Chen et al., 2024) which we use as our benchmarking suite. Section 5.1 of the main text shows results for the first four items in the following list.

- **Mean Shift.** To simulate systematic location bias, we perturb the posterior mean while keeping the covariance fixed: $q(\theta \mid y) = \mathcal{N}((1+\gamma)\mu_y, \Sigma_y)$. This mimics scenarios where the NPE model consistently misses the location of the true posterior mode.

- **Covariance Scaling.** To model over- or under-confidence in uncertainty quantification, we uniformly inflate (or deflate) the covariance matrix: $q(\theta \mid y) = \mathcal{N}(\mu_y, (1+\gamma)\Sigma_y)$. This captures calibration failures where the posterior has the correct shape and center but misrepresents its overall spread.

- **Anisotropic Covariance Perturbation.** We introduce structured distortion in the posterior shape by injecting uncertainty along the direction of least variance: $q(\theta \mid y) = \mathcal{N}(\mu_y, \Sigma_y + \gamma\Delta)$, where $\Delta = \mathbf{v}_{\min}\mathbf{v}_{\min}^\top$ and $\mathbf{v}_{\min}$ is the eigenvector of $\Sigma_y$ corresponding to its smallest eigenvalue. This subtly alters the posterior geometry.

- **Heavy-Tailed Perturbation.** To explore deviations in tail behavior, we replace the Gaussian with a multivariate $t$-distribution: $q(\theta \mid y) = t_\nu(\mu_y, \Sigma_y)$ where $\nu = 1/(\gamma+\epsilon)$. As $\gamma \to 0$, the distribution approaches Gaussian; increasing $\gamma$ yields heavier tails, modeling posterior approximations that spuriously introduce more extreme values.

- **Additional Mode.** We introduce a symmetric mode in the approximate posterior with weight $\gamma$, so that $q(\theta \mid y) = \gamma\mathcal{N}(-\mu_y, \Sigma_y) + (1-\gamma)\mathcal{N}(\mu_y, \Sigma_y)$ while $p(\theta \mid y) = \mathcal{N}(\mu_y, \Sigma_y)$. As $\gamma \to 0$, the two match, while increasing $\gamma$ increases the mass of the spurious mode.

- **Mode Collapse.** We introduce a symmetric mode in the true posterior with weight $\gamma$, so that $p(\theta \mid y) = \gamma\mathcal{N}(-\mu_y, \Sigma_y) + (1-\gamma)\mathcal{N}(\mu_y, \Sigma_y)$ while $q(\theta \mid y) = \mathcal{N}(\mu_y, \Sigma_y)$. As $\gamma \to 0$, the two match, while increasing $\gamma$ increases the mass of the missed mode.

Figures 6 to 11 present our experimental results, comparing our proposed methods against the baselines and including ablations on the number of calibration samples used in the conformal uniform test. Across all perturbation types, we observe consistent improvements over the classical C2ST under classifier degradation, highlighting the robustness of our method.

## B.2 Training Details

We use a three-layer neural network classifier with skip connections and a hidden dimension of 256. The network is initialized using PyTorch's default parameter initialization. Optimization is performed using the Adam optimizer with a cosine annealing learning rate schedule.

The training set consists of 2,000 samples: 1,000 labeled examples of the form $\{y_i, \theta_i, 1\}$, drawn from the joint distribution $p(\theta \mid y)p(y)$, and 1,000 negative examples from $q(\theta \mid y)p(y)$. Each pair $(\theta, y)$ lies in $\mathbb{R}^3 \times \mathbb{R}^3$. We train the model for 2,000 epochs using an initial learning rate of $1 \times 10^{-5}$, which is sufficient to ensure convergence in our experiments.

For training the DC classifier, we set $m = 10$, consistent with the configuration in its official repository. Increasing $m$ significantly prolongs training and makes the classifier harder to optimize effectively; we found $m = 10$ to be a practical sweet spot. The DC classifier is trained for 2000 epochs with a fixed learning rate of $1 \times 10^{-5}$

## B.3 High-Dimensional Image Experiments

We investigate the behavior of conformal two-sample testing (C2ST) methods under controlled image corruptions applied to CIFAR-10 data. The experimental pipeline consists of: (i) sampling "true" (uncorrupted) data from the empirical distribution, (ii) generating "fake" data via structured corruption operators parameterized by a strength parameter $\gamma$, (iii) training a discriminative classifier between the two sources under a conditional or unconditional formulation, (iv) constructing a family of interpolated "weak" classifiers via a parameter $\beta$ for robustness analysis, and (v) evaluating multiple conformal calibration strategies and a baseline C2ST $p$-value.

**Data Sampling and Corruption.**  Let $(\theta, y)$ denote (image, class label) pairs from the empirical dataset $D$ (CIFAR-10). In the conditional setting, $y$ is a class label and $\theta$ the corresponding image; in the unconditional setting only $\theta$ is used. We define:
$$(\theta, y) \sim p, \qquad (\tilde{\theta}_\gamma, y) \sim q_\gamma,$$
where the corrupted image $\tilde{\theta}_\gamma$ is obtain as $\tilde{\theta}_\gamma = \mathcal{C}_\gamma(\theta)$ where $\theta \sim p(\cdot \mid y)$ and $\mathcal{C}_\gamma$ is a corruption operator with strength $\gamma \geq 0$. The following corruption families are considered:

1. **Gaussian Blur** ("blur"): per-channel convolution with a Gaussian kernel of standard deviation $\sigma = \gamma$ (implemented via `scipy.ndimage.gaussian_filter`).

2. **Swirl Transformation** ("swirl"): a geometric warp (`skimage.transform.swirl`) with angular distortion parameter (strength) set to $\gamma$ and radius proportional to the image spatial extent.

3. **Additive Gaussian Noise** ("noise"): $\tilde{\theta}_\gamma = \theta + \epsilon$, where $\epsilon \sim \mathcal{N}(0, \gamma^2 I)$, followed by clipping to $[-3, 3]$ in normalized pixel space.

When $\gamma = 0$ the corruption reduces to the identity map. For each experiment we sample $N_{\text{true}} = 1024$ uncorrupted and $N_{\text{fake}} = 1024$ corrupted images to form the training pool. Additional independently sampled batches are used for evaluation replicates.

**Preprocessing.**  CIFAR-10 images are normalized channelwise to zero mean and unit (scaled) variance via the standard transform with mean $(0.5, 0.5, 0.5)$ and std $(0.5, 0.5, 0.5)$. All downstream embedding extraction resizes inputs to $299 \times 299$ (bilinear) and converts grayscale to 3-channels when necessary.

**Network Architecture.**  We employ a frozen Inception V3 backbone (pretrained on ImageNet) as a feature extractor producing a 2048-dimensional embedding $f(y) \in \mathbb{R}^{2048}$. In the conditional setting, a learnable label embedding $e(\theta) \in \mathbb{R}^{64}$ is concatenated to yield $[f(y); e(\theta)] \in \mathbb{R}^{2112}$. A feed-forward discriminator $g$ consists of:

$$\text{Linear}(d_{\text{in}}, H) \rightarrow \text{ReLU} \rightarrow \text{Dropout}(0.5) \rightarrow \text{Linear}(H, H/2) \rightarrow \text{ReLU} \rightarrow \text{Dropout}(0.5) \rightarrow \text{Linear}(H/2, 1),$$

with $H = 256$ (so hidden layers of sizes 256 and 128). The model outputs a logit $\ell = g(\cdot)$, trained with binary cross-entropy against labels 1 (true) and 0 (fake). Only the classifier head and (if used) label embeddings are trainable; the Inception backbone remains frozen.

**Weak Classifier Interpolation.**   To probe robustness and create a spectrum of discriminator strengths, we store (i) the randomly initialized classifier parameters $\psi_{\text{rand}}$ and (ii) the fully trained parameters $\psi_{\text{trained}}$. For any $\beta \in [0, 1]$, we define an interpolated ("weak") classifier:

$$\psi_\beta \;=\; (1 - \beta)\,\psi_{\text{trained}} \;+\; \beta\,\psi_{\text{rand}}.$$

Thus $\beta = 0$ recovers the trained discriminator and $\beta \to 1$ approaches a near-random classifier. We evaluate each $\beta$ independently in the downstream statistical tests.

**Training Procedure.**   Training is conducted for 200 epochs with Adam (learning rate $10^{-4}$) and cosine annealing LR scheduling. We employ Distributed Data Parallel (DDP) across all available GPUs. Instead of relying on a standard `DataLoader` with samplers, we materialize the full training set in CPU memory, deterministically shuffle each epoch (synchronized seeds across ranks), and partition indices among GPUs. Batch size per GPU is 256. This design avoids pinned-memory bottlenecks and enables explicit control of memory usage (with periodic cache clearing).

**Evaluation and Test Statistics.**   Let the (potentially interpolated) classifier yield logit $\ell(y)$ (conditional case notationally suppressed). We estimate:

- **Baseline C2ST $p$-value**: using the held-out evaluation samples, comparing score distributions between true and fake.

- **Conformal Multiple**: a conformal calibration procedure applied to embeddings (concatenated with label embeddings if conditional), producing $p_{\text{conf, mult}}$.

- **Conformal Uniform Tests**: scalability check by enlarging the reference (true) sample size by multiplicative factors $m \in \mathcal{M}$ (e.g., $\{2, 5, 20\}$), yielding $p_{\text{conf, uni}}^{(m)}$.

For each $(\gamma, \beta)$ configuration we perform $n_{\text{eval}}$ independent replicates (distinct random seeds), each resampling true/fake evaluation sets (default 64 runs unless otherwise specified). Success rates are reported as the empirical frequency of $p < \alpha$ with $\alpha = 0.05$ for each test variant.

**Hyperparameter Sweeps.**   We explore $\gamma$ ranges tailored to the perceptual sensitivity of each corruption type. Table 3 summarizes the grids and the shared $\beta$ values.

*Table 3.* Corruption strength grids $\Gamma$ per task and interpolation coefficients $\mathcal{B}$.

| Task | $\gamma$ (tested values) |
| --- | --- |
| Blur | $\{0.00, 0.36, 0.37, 0.38, 0.39\}$ |
| Swirl | $\{0.00, 0.25, 0.30, 0.35, 0.40, 0.50\}$ |
| Noise | $\{0.0, 0.001, 0.002, 0.003, 0.004, 0.005, 0.006, 0.007, 0.008\}$ |

$\mathcal{B} = \{0.0, 0.95\}$ (strong vs. weak discriminator regimes).

The narrow interval for blur emphasizes the phase transition region of detectability, while noise employs a single small variance ensuring subtle corruption. Swirl spans a broader geometric distortion spectrum.

## B.4   Evaluation Details

For evaluation, using the trained classifier, we sample $1000 \cdot m$ data points $\{y_i, \theta_i\}$ from $p(\theta \mid y)p(y)$, and another 1,000 samples from $q(\theta \mid y)p(y)$ to compute the rejection rate for C2ST and its conformal variants.

For classical C2ST and the Conformal Multiple testing variant, we set $m = 1$. For the Conformal Uniform test, we vary $m \in \{1, 2, 5, 10, 20, 50, 200\}$. We denote this setting as Conformal Uniform($m$), where $m$ specifies the evaluation sample budget.

For SBC and TARP, which require multiple posterior samples per observation $y$, we draw 200 posterior samples $\theta$ for each $y$. The overall evaluation budget remains consistent, using 1,000 pairs $(y, \theta)$ from $p(\theta \mid y)p(y)$. We denote these methods as SBC(200) and TARP(200) to indicate the number of posterior samples used. For DC, we also fix $m = 10$ and bootstrap 100 times to make sure it has sufficiently good power.

## C  Experiments on high dimensional posteriors with manifold structure

We further evaluate robustness and calibration of the proposed conformal tests in settings where the true posterior lies on a low-dimensional manifold embedded in a high-dimensional space. Specifically, we construct synthetic posteriors using a conditional normalizing flow model trained on data generated from a nonlinear spherical manifold, enabling controlled perturbations and precise comparisons across methods.

**Problem setup.**    We first sample $n$ points uniformly on the surface of the unit sphere in $\mathbb{R}^3$ using spherical coordinates $\phi \sim \mathcal{U}[0, \pi]$, $\tau \sim \mathcal{U}[0, 2\pi]$, and convert them to Cartesian coordinates:

$$a = \sin(\phi)\cos(\tau), \quad b = \sin(\phi)\sin(\tau), \quad c = \cos(\phi).$$

These coordinates are mapped into a higher-dimensional ambient space $\mathbb{R}^d$ (with $d = 100$) via a random projection matrix $W \in \mathbb{R}^{3 \times d}$ sampled from a standard Gaussian, yielding

$$\theta = W^\top (a, b, c)^\top + \varepsilon, \quad \varepsilon \sim \mathcal{N}(0, \sigma^2 I),$$

where $\theta \in \mathbb{R}^d$ and $\sigma > 0$ controls the noise level. The corresponding conditioning variable is defined as $y = (\phi/\pi, \tau/2\pi)$, so that $y \in [0, 1]^2$.

**Training details.**    We simulate posterior distributions using a conditional normalizing flow (CNF) model based on RealNVP (Dinh et al., 2016). Our architecture consists of 8 RealNVP layers, each parameterized by a neural network with one hidden layer of width 512. The model is trained for 500 epochs to ensure convergence. The true posterior is obtained by applying the inverse flow to a base sample $z \sim \mathcal{N}(\gamma, I)$, where $\gamma = 0$.

To simulate an approximate posterior, we perturb the base distribution by introducing a mean shift $\gamma \in \mathbb{R}^d$, such that $z \sim \mathcal{N}(\gamma, I)$ instead of the true $z \sim \mathcal{N}(0, I)$. The resulting approximate posterior $q(\theta \mid y)$ is thus generated by applying the same flow model to this mis-specified base distribution.

Figure 4 illustrates samples from the true and approximate posteriors for various values of $\gamma$, projected onto the first two principal components using Principal Component Analysis (PCA).

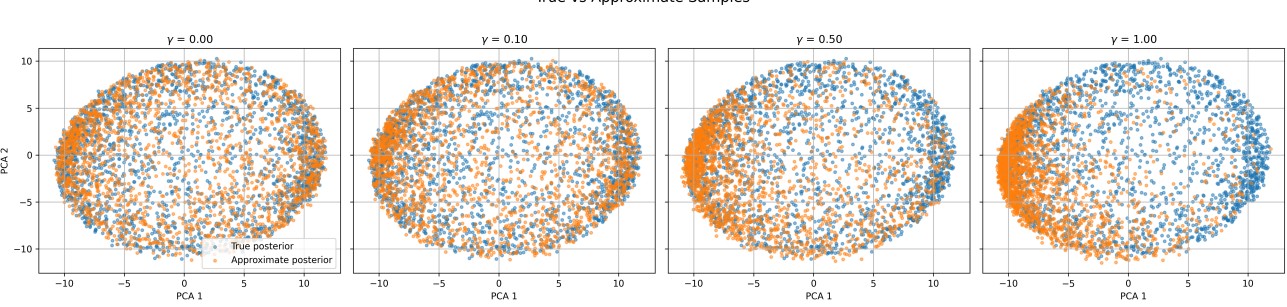

*Figure 4.* Approximate posterior samples (as a function of the base mean perturbation $\gamma$) projected on the first two principal axes

To distinguish between the true and approximate posteriors, we compute log density ratios by subtracting the log-probabilities assigned by the two CNF models. To emulate degradation in the classifier's discriminative ability, we add Gaussian noise $\delta \sim \mathcal{N}(0, \beta^2)$ to the log-density ratios. When $\beta = 0$, the scores are exact; as $\beta$ increases, the scores become progressively noisier, reflecting reduced discriminative power. Figure 5 shows that conformal variants of C2ST outperform all baselines under increasing posterior perturbation and retain a better test power under classifier degradation. Note that even though DC achieves good power in some settings, it failed to control the Type-I error rate, which is always the first priority when we conduct hypothesis testing.

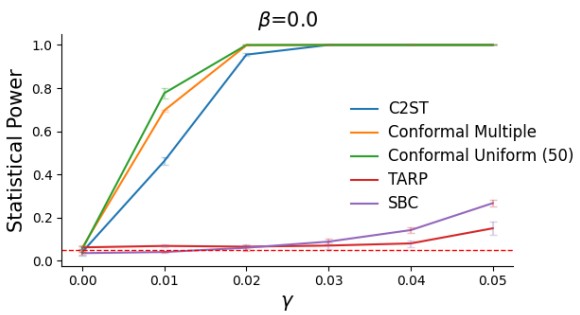 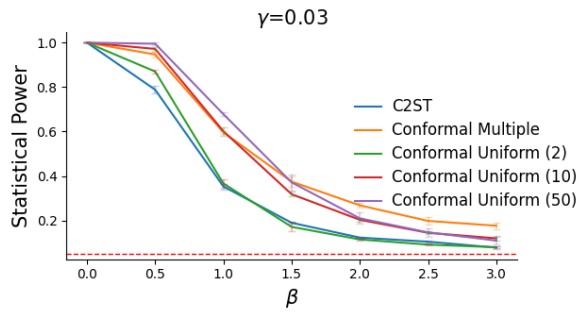

*(a)* Statistical power as a function of base mean perturbation $\gamma$     *(b)* Statistical power as a function of classifier noise $\beta$

*Figure 5.* Power analysis under mean perturbation of the base distribution of the normalizing flow model.

# D    A Practitioner's Guide

We collect here practical recommendations for applying the Conformal C2ST, distilling the theoretical and empirical findings of the main text into concrete guidance.

**Which test to use.**    The two variants trade simulation budget against power:

- **Drop-in replacement for C2ST $\to$ Conformal Multiple.** It uses a single shared calibration set from $p$, exactly matching the standard C2ST's simulation budget ($N$ samples each from $p$ and $q$) and inference cost. At that same budget it is already substantially more powerful and far more robust to weak classifiers than C2ST.

- **Maximum sensitivity when $p$-draws are cheap $\to$ Conformal Uniform$(m)$.** In NPE, sampling from the true joint $p(\theta, y) = \pi(\theta)\, p(y \mid \theta)$ typically does *not* require forward simulation through the learned model, so additional calibration from $p$ is inexpensive. Uniform$(m)$ draws a fresh calibration set per test point ($N \times m$ from $p$, still only $N$ from $q$), giving exact finite-sample validity (Lemma 3.1) and the highest power and resolution, especially for subtle discrepancies (small $\gamma$).

- **Expensive $p$-draws (e.g. high-fidelity simulators) $\to$ Multiple, or Uniform with small $m$.**

**Choosing the calibration size $m$ (Uniform).**    Power increases monotonically with $m$: the conformal $p$-value variance scales as $O(1/m)$ (Lemma A.3), so $p$-values concentrate on their non-uniform limit under the alternative more quickly. In our benchmarks most of this gain is realized by $m \approx 10$–50, with $m = 50$ a sensible default and little additional benefit beyond it (see the ablation in Appendix B.1). One caveat: under *severe* classifier degradation, fresh-calibration Uniform can fall below Multiple (e.g. Table 2, $\beta \geq 0.9$), since once scores are near-random the shared-calibration U-statistic aggregates the residual ranking signal more stably. Multiple is therefore a reasonable hedge in the very-weak-classifier regime.

**Choosing and training the classifier.**    The headline practical message is *not* "use any classifier." Rather: validity is automatic for any score (Lemma 3.1), but power is governed by *ranking quality* (AUC), not by thresholded accuracy or probability calibration. Concretely:

- Prefer higher-AUC scores or features; better ranking directly improves power. This is still far weaker than the standard C2ST requirement of a well-calibrated or near-Bayes-optimal classifier.

- Use the *continuous* score (classifier log-odds or signed margin), never the hard $0/1$ decision; any strictly increasing transform of the score gives an identical test, since the method depends only on ranks.

- Keep ties rare (Assumption 4.1): avoid heavily discretized, quantized, clipped, or saturated outputs, which create tie mass and erode the ranking signal.

- Weak, overfit, or off-the-shelf classifiers are fine provided they preserve some useful ordering. A frozen embedding with a light trainable head (as in our image experiments) is a practical default.

**Interpreting the output, and when the method cannot help.** Because Type-I control never depends on the classifier, a weak classifier yields a *conservative but valid* test, never an inflated one—so a model that "passes" is more trustworthy here than under C2ST, where a pass can simply reflect low power. Two limits are worth keeping in mind. First, the test is *global*: it certifies joint validity of $q$ but does not localize where $q$ deviates from $p$; pair it with a local diagnostic if localization is needed. Second, there is a hard floor: if the score carries no ordering information ($\text{AUC} \approx \frac{1}{2}$—an almost-constant score, or $p$ and $q$ nearly identical or flat over the relevant region), then no ranking-based test, ours included, can recover power, and the test simply sits at the nominal level.

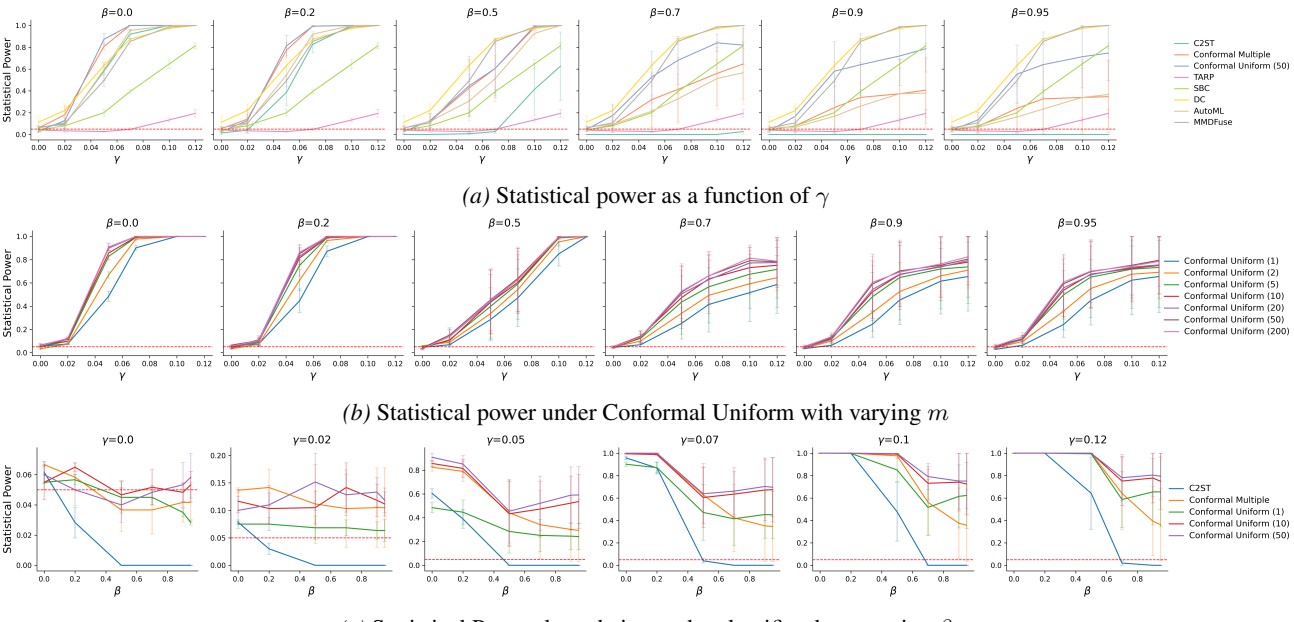

*(a)* Statistical power as a function of $\gamma$

*(b)* Statistical power under Conformal Uniform with varying $m$

*(c)* Statistical Power degradation under classifier degeneration $\beta$

*Figure 6.* Power analysis under Mean Shift

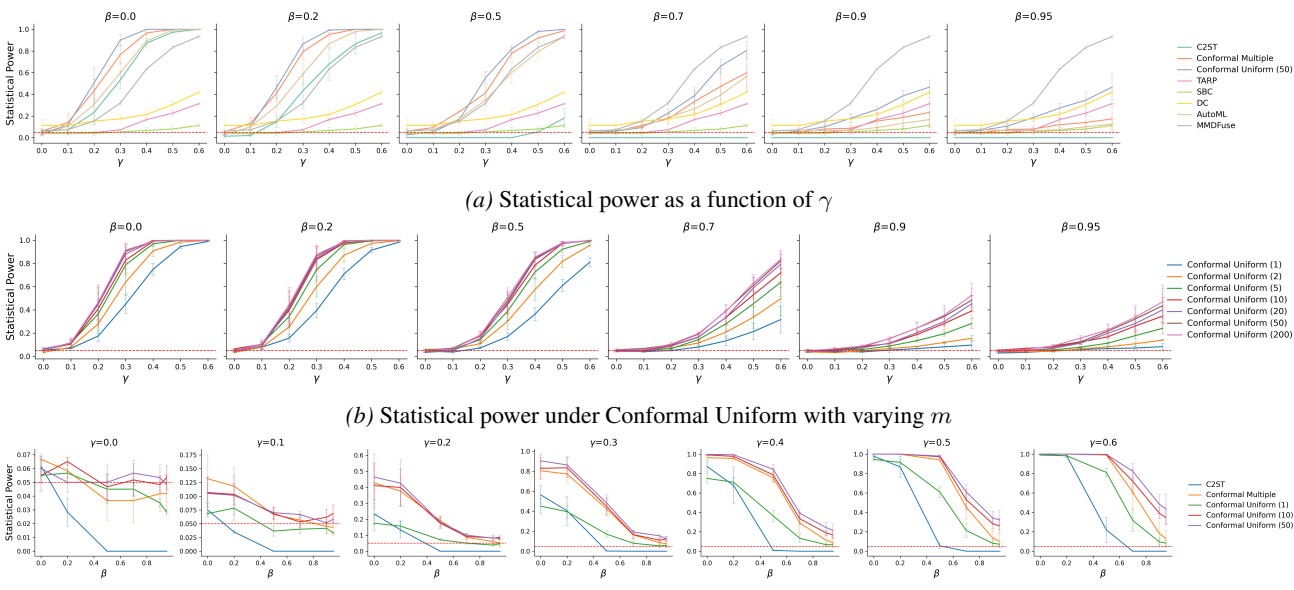

*(a)* Statistical power as a function of $\gamma$

*(b)* Statistical power under Conformal Uniform with varying $m$

*(c)* Statistical Power degradation under classifier degeneration

*Figure 7.* Power analysis under Covariance Scaling.

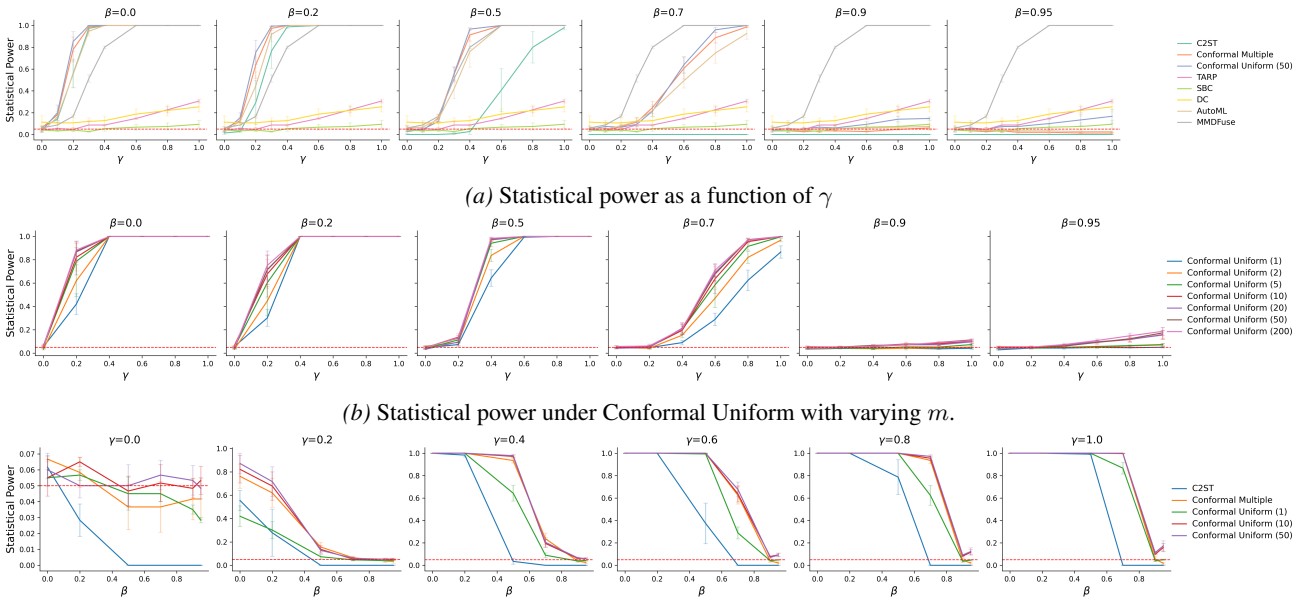

*(a)* Statistical power as a function of $\gamma$

*(b)* Statistical power under Conformal Uniform with varying $m$.

*(c)* Statistical Power degradation under classifier degeneration

*Figure 8.* Power analysis under Anisotropic Covariance Perturbation.

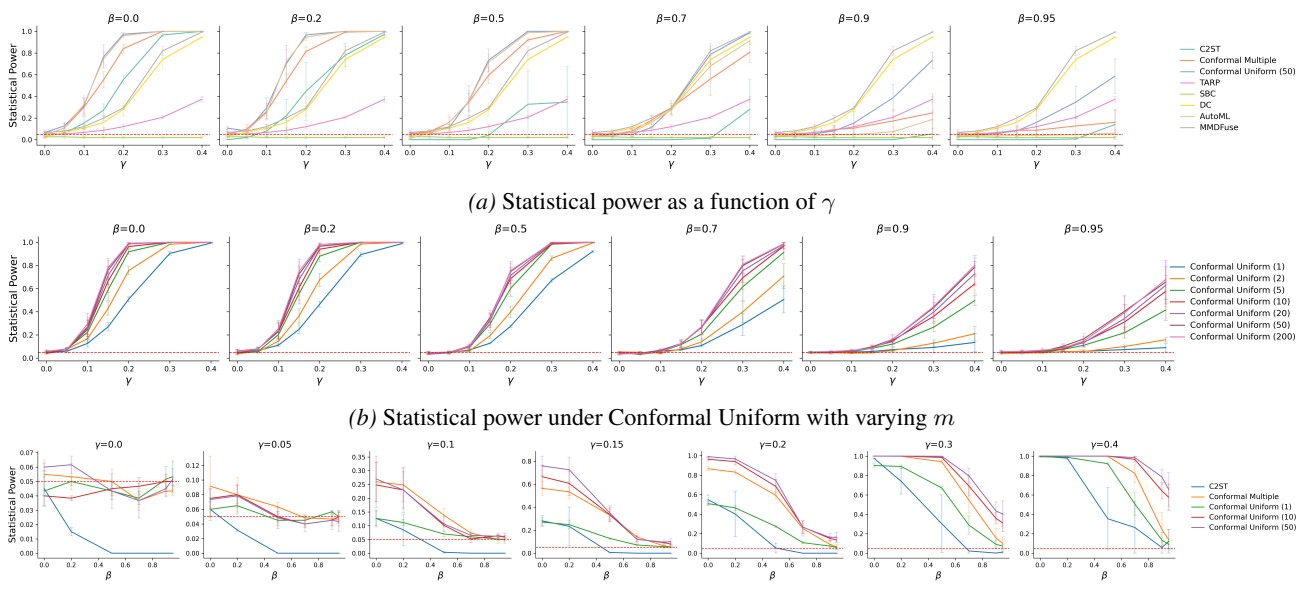

*(a)* Statistical power as a function of $\gamma$

*(b)* Statistical power under Conformal Uniform with varying $m$

*(c)* Statistical Power degradation under classifier degeneration

*Figure 9.* Power analysis under Heavy-Tailed Perturbation.

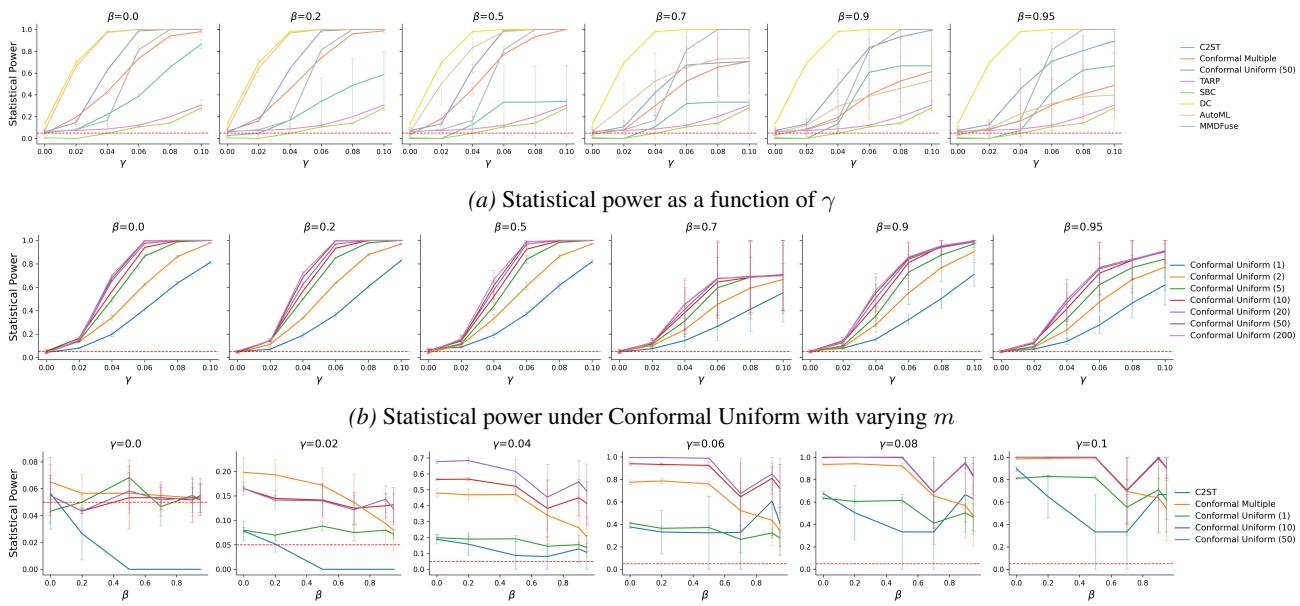

*(a)* Statistical power as a function of $\gamma$

*(b)* Statistical power under Conformal Uniform with varying $m$

*(c)* Statistical Power degradation under classifier degeneration

*Figure 10.* Power analysis under Additional Mode.

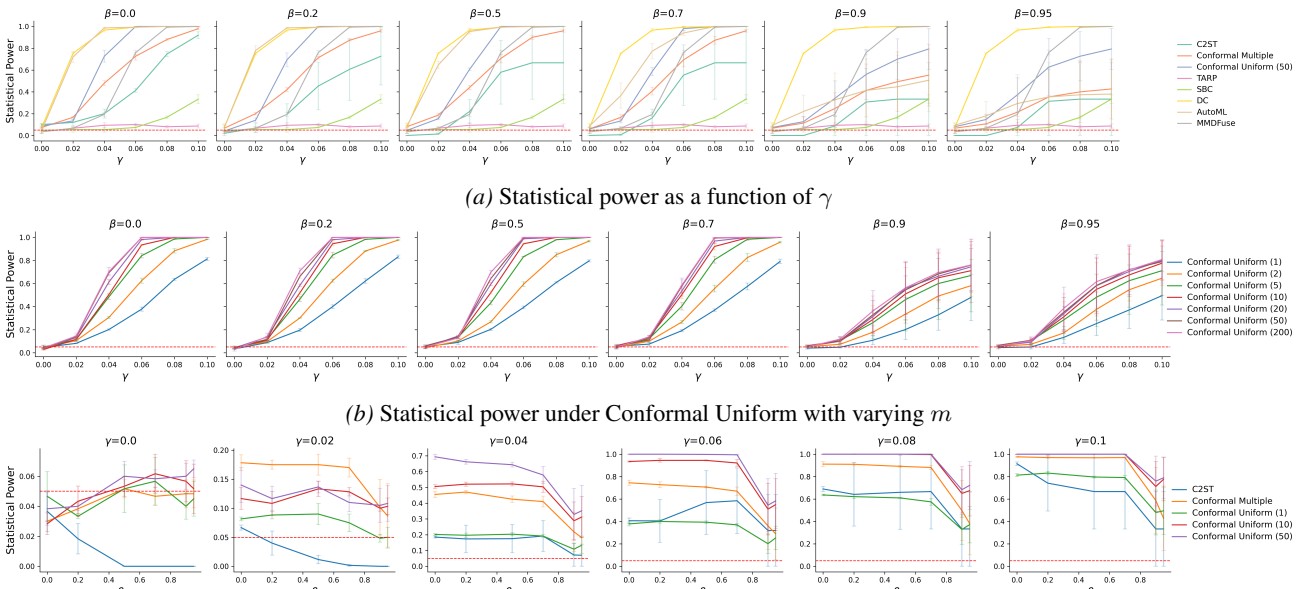

*(a)* Statistical power as a function of $\gamma$

*(b)* Statistical power under Conformal Uniform with varying $m$

*(c)* Statistical Power degradation under classifier degeneration

*Figure 11.* Power analysis under Mode Collapse.

