# OpenReview forum: "Conformal C2ST: Turning weak classifiers into strong two-sample tests"
_ICML.cc/2026/Conference — ICML 2026 regular_

### Official Review · Reviewer_t9Vt · 2026-03-08

**Soundness:** 4
**Presentation:** 3
**Significance:** 3
**Originality:** 3
**Overall Recommendation:** 4
**Confidence:** 3

**Summary:**

This paper applies a conformalized classifier-based two-sample test (C2ST) to the issue of diagnosing the accuracy of neural posterior estimation (NPE). While the protection of Type I error through conformalization is well-understood, this work demonstrates that the conformalized C2ST still has good power even under classifier degradation. They justify this with theory and practical demonstration, which demonstrates the improved power of their method compared to other methods in the literature.

**Compliance With Llm Reviewing Policy:**

Affirmed.

**Final Justification:**

The authors have clarified the difference in scope and extensions beyond Hu & Lei (2024), and I have maintained my score due to methodological overlap with Hu & Lei (2024). Nonetheless, this is an interesting application of the method, and the focus on robustness is interesting.

**Key Questions For Authors:**

1. In general, I think it would be helpful to clarify the differences to Hu & Lei (2024) more carefully. In the "Summary of contributions" section, the authors state that "Their method fundamentally relies on a nearly-Bayes-optimal classifier to model the density ratio between distributions". I found this comment confusing, as I thought the approach of Hu & Lei (2024) relied on the same conformity score as (2)?
Are the main differences of this work the considerations of power and the simpler KS uniform test which is made possible due to the NPE context and access to infinite samples from $p$?

2. Your testing procedure is actually between the joint densities $p(\theta,y)$ and $q(\theta,y)$. While they both have the same marginal distribution $\pi(y)$, would this pose any challenges to detecting the differences between $p(\theta \mid y)$ and $q(\theta \mid y)$ if $y$ is high-dimensional and noisy? I just wonder if noise in $y$ will swamp the signal for the posteriors.

3. I found the discussion surrounding Lemma 2.3 a little bit meandering. Is the main point of this section to justify the choice of (2) as it gives the nice form of $E[U]$ under the alternative in Lemma 2.3? I wonder if this explanation can be streamlined a bit, as it was a bit confusing.

**Limitations:**

Yes.

**Strengths And Weaknesses:**

Strengths: The paper is quite well-written, and provides a comprehensive theoretical and empirical understanding of the conformalized C2ST within their NPE framework. The theory is thorough and sound to my understanding, and the experimental results convincingly demonstrate the usefulness of their method within the context of NPE.

Weaknesses: I found certain sections a little bit dense and unclear to read, for example the area surrounding the AUC discussion.
The method is also arguably an application of the work of Hu & Lei (2024), although there may be some differences that can be emphasized better. I elaborate on this in my questions below.

---

> ### Author Rebuttal · Authors · 2026-03-31
>
> Thank you for the careful and encouraging review. We appreciate your positive assessment of the theory and experiments. Your questions are especially helpful, and we will make the distinction from Hu & Lei and the role of the AUC discussion much clearer in the revision.
>
> ### 1) Difference from Hu & Lei (2024)
>
> Hu & Lei do use the analogous *oracle* conformity score in their setting—the conditional density ratio—and they also note that only the induced ranking matters, i.e. the score need only be estimated up to a monotone transform. So the key distinction is **not** a different score function.
>
> Rather, the papers address different goals. Hu & Lei develop a general conformal test for conditional two-sample testing with possibly *unequal* covariate marginals, using weighted conformalization and a weighted rank-sum / U-statistic. In our NPE setting, the relevant marginals are equal by construction, so that weighting correction is absent; our shared-calibration multiple test can be viewed as a simplified specialization of their statistic.
>
> What is new here is twofold:
> (i) in this equal-marginal setting we can additionally construct an exact finite-sample **uniformity** test based on independently calibrated conformal p-values, while our shared-calibration multiple test is a simplified specialization of Hu & Lei’s statistic; and, more importantly,
> (ii) we provide a **robustness theory for weak plug-in classifiers** that is not present there. Our Theorems 2.5 and 2.7 quantify how conformal power degrades under score-estimation error, and make the signal directly interpretable in AUC / ranking terms. In particular, the degradation is governed by **misranking / AUC loss** rather than by hard-threshold classification accuracy or calibrated probability estimation. We will revise the paper to make this distinction precise.
>
> A related assumptions point is also worth stating explicitly. Hu & Lei’s local-alternative analysis assumes the oracle score itself has a continuous distribution with bounded density. Our Assumption 2.4 is weaker and more targeted to the ranking problem: it only requires bounded density **near zero** for the pairwise difference $r(\tilde X)-r(\tilde X')$, i.e. a low-noise condition on near-ties in the ordering. This is exactly the comparison we should have made more clearly in the original draft.
>
> ### 2) Does high-dimensional / noisy $y$ swamp the signal?
>
> This is a very good question. In principle, yes: if $y$ is extremely high-dimensional and mostly irrelevant, then learning a useful score on $(\theta,y)$ becomes harder. But this is precisely where standard C2ST can be brittle, and why conformalization helps. C2ST uses a hard threshold at $0.5$, whereas conformal C2ST aggregates **weak ranking information** across many draws. So even when the classifier is far from ideal in the full joint space, it can still produce informative orderings that the conformal test converts into power.
>
> We now have stronger empirical evidence for this point. In the matched-budget experiments added for Reviewer KQwH, the conformal methods remain far more robust than standard C2ST under severe classifier degradation. We also added a realistic **gravitational-lensing** benchmark (please see response to Reviewer 1c8u), where $(\theta,y)$ is high-dimensional and structured: the conformal tests stay calibrated on the exact sampler and detect the biased sampler earlier than AutoML/C2ST. We will mention these new results in the revision to better address this concern.
>
> Of course, if the score becomes essentially uninformative, no method can recover power. But our message is that conformal C2ST remains useful under much weaker conditions than standard C2ST because it only needs some nontrivial ranking signal, not a highly accurate thresholded classifier.
>
> ### 3) Clarifying the role of Lemma 2.3 / the AUC discussion
>
> We agree that this part can be streamlined. The main purpose of Lemma 2.3 is to explain why the density ratio $r=p/q$ is the right oracle score for conformal testing: it maximizes AUC, and the expected conformal p-value under the alternative satisfies
> $$
> \mathbb E[U]=1-\mathrm{AUC}(r).
> $$
> So better ranking directly means more skew toward small p-values and hence more power. In other words, the point of this section is not only to justify the score in (2), but to connect **oracle ranking quality** to the mechanism by which conformal p-values deviate from uniformity under the alternative.
>
> We will rewrite this discussion more directly around that message. We will also move the AUC / ranking interpretation of our robustness results forward in the main text, since this is really the central conceptual takeaway: conformal C2ST succeeds when the score preserves useful orderings, even if the classifier is weak, biased, or poorly calibrated.
>
> Thank you again for these suggestions. We believe the revision will make the relationship to Hu & Lei, the role of noisy $y$, and the purpose of the AUC discussion much clearer.

---

> > ### Author Rebuttal · Reviewer_t9Vt · 2026-04-01
> >
> > Thank you for addressing all of my comments clearly and thoroughly. While I appreciate the clarification of the difference in scope and extensions beyond Hu & Lei (2024), I believe the methodology can still be primarily viewed as an application of the conformal score of Hu & Lei (2024) - I will thus maintain my score.

---

> > > ### Author Response · Authors · 2026-04-02
> > >
> > > Thank you for the careful follow-up and for engaging so thoughtfully with the comparison to Hu & Lei (2024).
> > >
> > > We understand your view that the conformal backbone is inherited. However, our intended claim is more specific: in the equal-marginal NPE setting, this framework yields an exact finite-sample Uniform variant, a new weak-classifier robustness/power analysis, and strong empirical gains in posterior validation, including under matched-budget comparisons and classifier degradation.
> > >
> > > We appreciate your positive assessment of the theory and experiments, and will revise the framing to make this distinction as clear and precise as possible.

---

### Official Review · Reviewer_n84m · 2026-03-09

**Soundness:** 3
**Presentation:** 2
**Significance:** 3
**Originality:** 3
**Overall Recommendation:** 4
**Confidence:** 3

**Summary:**

The manuscript studies classifier-based two samples (C2ST) tests for evaluating approximate Bayesian inference methods, specifically neural posterior estimation (NPE, were C2ST is commonly used for benchmarks), with a focus on scenarios where the underlying classifier is weak. The manuscript proposes **conformal C2ST**  conformalized version of C2ST that turn (e.g. weak) classifier scores into exact finite-sample p-values. Theoretically it is shown that the derive p-value satisfies uniformity under null ($p=q$) for any scoring rule (i.e .also weak classifiers i.e. from standard conformal prediction arguments) and that the *oracle* scoring function for power is the density ratio (i.e. which can be obtained from an Bayes optimal classifier). In addition it is shown that under the alternative ($p\neq q$) the conformal p-values are systematically smaller than uniform (given the oracle scoring rule) and **importantly** it is also show that this result is robust to suboptimal scoring rules (i.e. weak classifiers). The manuscript then verifies these findings with empirical results on controlled benchmarks (classifier is artificially degraded in certain ways) as well as a higher dimensional experiment with images + blue degradations, which do show improved performance.

**Compliance With Llm Reviewing Policy:**

Affirmed.

**Final Justification:**

The authors have addressed my concerns and answered my question clearly. I hence keep my positive evaluation.

**Key Questions For Authors:**

The robustness results are intesting but formulated somewhat indirect i.e. controlled via L2 error to the density version. This is kinda hard to interpret/judge practically. Is there a version of the robustness theory stated directly in terms of ranking error, AUC gap, Accuracy, or pairwise misordering probability?
- Can the authors better separate what comes from conformal calibration itself versus what comes from extra calibration samples from p, especially for Conformal Uniform? (i.e. reduce N, by M).
- While this work mostly adresses C2ST as a test for $q=p$ and its statistical efficiency to accept/reject this null hypothesis. The C2ST accuracy  is also often used as a "metric" to compare different models (effectively C2ST acc = (1+TV(p,q))/2). I wonder if conformal C2ST can also be used to estimate this metric more reliably, as a current limitation is that C2ST accuracy estimation is biased and depends strongly on the classifier.

**Limitations:**

The limitations are clearly stated in the corresponding section.

**Strengths And Weaknesses:**

Strengths:
- Well written/structured, all claims/lemmas are formally stated and proven in the Appendix. The proposed method via conformalization is elegant and practically simple. Validity comes from conformal exchangeability, not classifier quality; power comes from ranking ability, not hard 0.5-threshold accuracy (which even weak classifiers can do to some degree).
- Meaningful theoretical contributions: The finite-sample + robustness theorems are to my knowledge novel and interesting contributions. The numerical experiments are well designed to support the claims further.

Weaknesses
- The strongest guarantee applies only to the "uniform" variant i.e. exact finite-sample validity. But for the shared-calibration "multiple" variant the guarantees appear asymptotic rather than finite-sample (or?). Since the multiple variant is more computationally comparable to standard C2ST, I think the paper should be more explicit about this distinction in the main text and temper the framing accordingly.
- Fairness of experimental comparisons is somewhat mixed. The conformal uniform tests is allowed to use many more samples from p than standard C2ST and some baselines. I would like a clearer matched-budget comparisons (specifically because samples from p are usually the more expensive ones). Fig 3 (image experiment) is not fully convincing
- Minor: Presentation can be improved:
	- Figure 3 its kinda hard to find out what line correspond to what labels. The fontsize is also way to small.
	- Title is somewhat confusing i.e. the paper does not show that C2ST is strong with weak classifiers but instead proposes a method i.e. (conformal C2ST) that is.
	- Some typos i.e. Line 12 "A fundamental problems"

---

> ### Author Rebuttal · Authors · 2026-03-31
>
> Thank you for the thoughtful and encouraging review. You are correct: validity comes from conformal exchangeability, while power comes from ranking ability rather than hard $0.5$-threshold accuracy.
>
> ### 1) Exact vs. asymptotic guarantees
>
> The strongest finite-sample guarantee is for Uniform; Multiple (shared calibration) is asymptotically valid. Multiple is the directly matched-complexity analogue of standard C2ST: one shared calibration set, essentially the same inference cost. So they are complementary: Uniform = exact finite-sample validity, higher power when extra $p$-samples are cheap; Multiple = asymptotically valid, budget-matched to C2ST.
>
> ### 2) Matched-budget fairness and what comes from conformalization vs. extra samples
>
> There are two gains: (i) conformalization, replacing thresholded accuracy by rank-based aggregation; (ii) the stronger Uniform design, using fresh $p$-calibration for each test point.
>
> Multiple reuses one calibration set, induces dependence, and therefore needs the U-statistic / variance-normalized construction. So Uniform is not merely lower-variance Multiple; it is an inherently different test with stronger calibration structure.
>
> Hence the right comparisons are Conformal-Multiple vs. C2ST (isolates conformalization at matched budget) and Conformal-Uniform vs. Conformal-Multiple (isolates fresh independent calibration + larger effective $m$).
>
> Appendix B.1 includes an ablation over $m$ for Conformal-Uniform, showing that increasing $m$ improves power, echoing Lemma A.3. Figure 2a and our response to Reviewer KQwH and Reviewer 1c8u further separate the two effects. Even under matched-budget Multiple, conformalization alone already yields a large gain over C2ST under classifier degradation:
> distorted variance $(\gamma=.3,\beta=.5)$ $0.67\to55.33$; perturbed mean $(\gamma=.07,\beta=.5)$ $2.67\to60.67$; perturbed variance $(\gamma=.4,\beta=.5)$ $0.00\to78.00$; tail $(\gamma=.3,\beta=.5)$ $32.67\to92.00$.
>
> So Conformal-Multiple answers the fairness question and shows that conformalization itself gives a substantial gain over C2ST. Moreover, Conformal-Uniform is stronger for two reasons—larger effective calibration and fresh independent calibration—adding a further gain *over Conformal-Multiple* when simulator draws from $p$ are cheap: distorted variance at $\gamma=.2$: $78.00\to85.33$; perturbed variance at $\gamma=.3$: $76.67\to90.00$.
>
> We will rewrite this part of the paper accordingly and make the distinction between these two gains much clearer.
>
> ### 3) Robustness theory in terms of AUC / ranking error
>
> Our robustness theory is fundamentally about ranking quality, not calibrated density-ratio estimation per se. The $L_2$ assumption is an upstream condition on score-estimation error, sufficient to derive a direct statement in terms of AUC / pairwise ranking error. For Multiple, this is already built into Theorem 2.7 through $\Delta_s:=\mathrm{AUC}(s)-\frac12$: the deterministic drift is exactly this AUC signal, and the plug-in/oracle relative gap is $O_p(\varepsilon^{2/3})$. For Uniform, if $U_s$ is the conformal p-value from a fixed deterministic score $s$, then $\mathbb E[U_s]=\frac{m}{m+1}(1-\mathrm{AUC}(s))+\frac{1}{2(m+1)}$, hence the oracle--plug-in gap in expected conformal p-value is exactly the AUC gap, up to the finite-$m$ factor. Thus a weak classifier remains useful precisely when it preserves nontrivial pairwise orderings, even if its probabilities are poorly calibrated or its $0.5$-threshold accuracy is poor.
>
> ### 4) Can conformal C2ST be used as a discrepancy metric?
>
> We distinguish between finite-sample classifier accuracy and Bayes accuracy with equal class priors, $(1+\mathrm{TV}(p,q))/2$: only the latter is directly tied to total variation. In our setting, conformalization suggests a more robust discrepancy quantity based on score rankings rather than hard decisions. For the multiple test, the natural effect size is $\widehat\Delta_{\mathrm{conf}}:=\frac12-\frac{1}{n_q}\sum_{j=1}^{n_q}\widehat U_j$, with population target $\Delta_{\mathrm{conf}}(s)=\mathrm{AUC}(s)-\frac12$. Also, $\mathbb E[U]=1-\mathrm{AUC}(r)$ and $\mathrm{AUC}(r)\ge \frac{1+\mathrm{TV}(p,q)}{2}$, so $\Delta_{\mathrm{conf}}(r)\ge \frac{\mathrm{TV}(p,q)}{2}$. This is not an exact estimator of TV, but it is directly linked to separability, avoids dependence on a fixed threshold, and remains stable under imperfect score estimation. We will present $1/2-\bar U$ as a discrepancy metric complementary to hypothesis testing.
>
> ### 5) Presentation fixes
>
> In the revision we will make the Uniform vs. Multiple distinction explicit earlier, enlarge and simplify Figure 3, revise the title to foreground Conformal C2ST more clearly, and fix typos such as “A fundamental problems.
>
> Overall, your review helped us sharpen the paper’s main message. We believe the added matched-budget experiments, failure-mode discussion, and clearer AUC-based interpretation substantially strengthen that case.

---

> > ### Author Rebuttal · Reviewer_n84m · 2026-04-03
> >
> > I thank the authors for their detailed response. My concerns have been resolved. I will keep my already positive evaluation of the manuscript.

---

> > > ### Author Response · Authors · 2026-04-04
> > >
> > > Thank you again for reading the rebuttal carefully. We appreciate your note that the concerns were fully resolved. If there are any remaining technical points, clarifications, or presentation improvements that would be useful for the discussion, we would be very happy to address them.

---

### Official Review · Reviewer_1c8u · 2026-03-13

**Soundness:** 3
**Presentation:** 3
**Significance:** 2
**Originality:** 3
**Overall Recommendation:** 4
**Confidence:** 3

**Summary:**

This paper investigates the challenge of validating neural posterior estimation (NPE) in simulation-based Bayesian inference, where the goal is to determine whether a learned posterior distribution truly matches the intractable ground truth. Traditional approaches such as the Classifier Two-Sample Test (C2ST) rely on strong classifiers to distinguish between samples from the true and approximate joint distributions, but their effectiveness diminishes when classifiers are weak, biased, or overfit.

The authors propose Conformal C2ST, a framework that leverages conformal calibration to transform arbitrary classifier scores into exact finite-sample p-values. This approach guarantees valid hypothesis testing regardless of classifier quality, while still retaining non-trivial statistical power under alternatives. Theoretical results establish finite-sample Type I error control and robustness to classifier estimation error, with power degrading gracefully as classifier quality worsens.

Empirically, the method is evaluated across a range of controlled perturbations to posterior distributions. Results show that Conformal C2ST consistently outperforms classical C2ST, Simulation-Based Calibration (SBC), and TARP, particularly in scenarios where classifiers are weak or mis-specified. The framework offers both a uniform test and a multiple/shared calibration test, balancing power and sample efficiency.

Overall, the paper demonstrates that weak classifiers, when combined with conformal calibration, can serve as surprisingly strong tools for posterior validation, providing a robust and practical diagnostic for modern simulation-based inference workflows.

**Compliance With Llm Reviewing Policy:**

Affirmed.

**Final Justification:**

This paper addresses a highly significant challenge in simulation-based inference: validating neural posterior estimates when the available classifiers are weak, biased, or overfit. The authors propose Conformal C2ST, an elegant and original framework that integrates conformal calibration with classifier-based two-sample testing to provide robust, finite-sample valid p-values.

Strengths & Weaknesses (Initial Assessment):
Initially, I found the paper to be methodologically rigorous and technically sound, with clear presentation and a strong conceptual contribution. The theoretical guarantees surrounding Type I error control and robustness to classifier estimation error are compelling. However, my primary concerns centered on the lack of real-world scientific case studies, the somewhat abstract nature of the theoretical assumptions in high-dimensional spaces, and a need for more explicit practical guidelines regarding computational trade-offs and comparisons to newer local diagnostic methods.

Rebuttal Evaluation:
The authors submitted a thorough and highly constructive rebuttal that successfully addressed my main concerns:

Real-world validation: The addition of the gravitational lensing case study is highly commendable. It convincingly demonstrates the utility of Conformal C2ST in a practical, physics-based inverse problem and clearly shows its superiority in detecting subtle biases compared to classical C2ST and TARP. This significantly strengthens the paper's practical significance.

Practical Guidelines: The explicit heuristics provided for practitioners (e.g., when to choose Conformal-Multiple for fixed budgets vs. Conformal-Uniform for higher sensitivity) add substantial value and directly address my questions about computational budget trade-offs.

Assumptions and Comparisons: In the discussion phase, the authors provided much-needed clarity on the failure modes of Assumptions 2.4 and 2.6 (e.g., highlighting issues with flat posteriors or highly discretized learned scores). Furthermore, their explicit contrast between the global nature of Conformal C2ST and the localized focus of recent posterior reshaping methods effectively positions this work within the broader diagnostic landscape.

Final Recommendation:
The rebuttal reinforced my positive prior assessment and clarified my remaining reservations. The authors have demonstrated that their method is not just theoretically sound, but practically applicable and robust in realistic scientific workflows.

I maintain my recommendation of **Weak Accept** (4) (leaning strongly positive). This is a technically solid paper that introduces a highly useful and creative paradigm shift for NPE validation. I strongly encourage the authors to seamlessly incorporate the new gravitational lensing tables, the concrete examples of assumption failure modes, and the expanded limitations discussion (global vs. local diagnostics) into the camera-ready version.

**Key Questions For Authors:**

Assumption realism: How realistic are Assumptions 1–2 in high-dimensional scientific models? Could you provide empirical evidence or discussion of when these assumptions might fail?

Sample efficiency: Can you provide a more systematic analysis of power vs. calibration size 𝑚 and computational budget, to guide practitioners in choosing between Uniform and Multiple variants?

Real-world case studies: Could you demonstrate Conformal C2ST on real NPE applications (e.g., astrophysics, genetics) to validate its utility beyond synthetic benchmarks?

Comparison with recent diagnostics: How does Conformal C2ST compare with newer posterior diagnostic methods (e.g., local posterior reshaping) on shared benchmarks?

Practical adoption: Could you provide guidelines or heuristics for practitioners on how to select classifiers and calibration strategies in typical NPE workflows?

**Limitations:**

No. The paper does not adequately discuss limitations and potential negative societal impact. While it acknowledges assumptions and focuses on synthetic experiments, it does not sufficiently address: The realism of theoretical assumptions in complex, high-dimensional models. The lack of real-world case studies, which limits immediate applicability. Computational efficiency trade-offs, which are important for practitioners with limited resources.

Constructive suggestions:Expand discussion on the realism of assumptions and potential failure modes. Include at least one real-world case study to demonstrate applicability. Provide a systematic analysis of sample efficiency and computational cost. Discuss broader societal implications of robust posterior validation, especially in scientific domains where incorrect inference could mislead research.

**Strengths And Weaknesses:**

Soundness
The paper is technically sound and methodologically rigorous. The proposed Conformal C2ST framework is carefully defined, with strong theoretical guarantees: exact finite-sample Type I error control and quantified robustness to classifier estimation error. The theoretical analysis is clear, with lemmas and theorems that connect power to AUC and TV distance, and robustness bounds scaling as
𝑂(𝜀2/3). The experiments are well designed, particularly the classifier degradation studies that directly test the central claim. Weaknesses include somewhat idealized assumptions (e.g., bounded density near zero, MSE bounds) whose realism in complex scientific models is not fully examined, and limited exploration of computational efficiency trade-offs.

Presentation
The paper is clearly written, well structured, and easy to follow. The narrative flows logically from problem setup to theoretical contributions to empirical validation. The exposition of classical C2ST limitations and the motivation for conformal calibration is particularly clear. The proofs and assumptions are stated carefully, though a deeper discussion of their practical realism would improve clarity. The experiments are presented systematically, but the lack of real-world case studies leaves the reader wanting more context on applicability.

Significance
The work addresses an important problem in simulation-based inference: validating neural posterior estimates. Posterior validation is critical for scientific applications in astrophysics, genetics, and other domains where simulation-based inference is widely used. By showing that weak classifiers can still yield valid and powerful tests when combined with conformal calibration, the paper advances practical diagnostic tools for NPE workflows. The significance is high, as the method is easy to adopt and could become a standard diagnostic. The impact would be even greater with demonstrations on real-world scientific datasets.

Originality
The originality lies in the novel combination of conformal calibration with classifier-based two-sample testing. While conformal inference is not new, its integration into the C2ST paradigm for posterior validation is creative and impactful. The insight that classifier ranking information, rather than accuracy, suffices when combined with conformal calibration is conceptually important. This reframing makes C2ST robust to weak classifiers, which is highly relevant in practice. The originality is thus in the methodological upgrade and its application to NPE validation, rather than in inventing a new theoretical tool from scratch.

---

> ### Author Rebuttal · Authors · 2026-03-31
>
> Thank you for the careful and encouraging review. We appreciate your positive assessment of the paper’s technical soundness and relevance. We agree the revision should better explain assumption realism, budget tradeoffs, and practical guidance, and we have also added a real NPE case study.  We will also broaden the limitations/impact discussion: incorrect posterior validation can mislead scientific conclusions, and our method does not address simulator misspecification (the M-open problem).
>
> ### 1) Assumption realism
>
> We will expand the discussion of when the assumptions may fail. Assumption 2.4 is a pairwise low-noise condition on $Z=r(\tilde X)-r(\tilde X')$: it rules out degenerate settings where the density ratio is locally flat near the ranking boundary. Assumption 2.6 is a rare-ties condition, which is mild for continuous neural classifier outputs. We will expand the revision to make both the scope and the failure modes explicit.
>
> Importantly, we should **not** equate AUC$=1/2$ with “no information.” AUC$=1/2$ can occur for non-constant scores, and such scores may still yield non-uniform conformal p-values under the alternative. The truly degenerate case is a constant (or nearly constant) score, which destroys ordering entirely. At the same time, our theory and experiments do show that power depends critically on ranking quality, so higher AUC is still preferred. The key point is that this is a much weaker requirement than needing a well-calibrated or near-Bayes-optimal classifier: conformal C2ST only needs useful ranking signal, not accurate probability estimates.
>
> ### 2) Real-world study: gravitational lensing
>
> We have now added a real NPE case study based on a gravitational-lensing benchmark closely following TARP (Lemos et al., 2023, App. E), rather than designing a custom scientific task. Concretely, we fit a Gaussian prior over $16\times16$ source images from galaxy data, use a linear lensing-style forward model
> $$y=A\theta+\varepsilon,$$
> with affine warp + Gaussian blur + additive noise, and compare reverse-SDE samplers driven by (i) the exact conditional score and (ii) the biased approximation used in TARP. We also interpolate between these with
> $$\nabla_\theta \log p_{t,\gamma}(x\mid\theta)
> =(1-\gamma)\nabla_\theta \log p_t(x\mid\theta)+\gamma\nabla_\theta \log \hat p_t(x\mid\theta).$$
>
> | Setting | SBC | TARP | AutoML | C2ST | Conf. Mult. | Conf. Unif. (50) |
> |---|---:|---:|---:|---:|---:|---:|
> | Exact sampler ($\gamma=0$) | 10 | 4 | 4 | 2 | 2 | **5** |
> | Interpolated bias ($\gamma=0.6$) | 6 | 6 | 28 | 28 | 50 | **66** |
> | Interpolated bias ($\gamma=0.7$) | 6 | 6 | 70 | 76 | 92 | **98** |
> | Fully biased sampler ($\gamma=1.0$) | 6 | 24 | 100 | 100 | 100 | **100** |
>
> These results are important because they show the desired behavior in a realistic inverse problem: the conformal tests stay calibrated when the posterior sampler is correct and detect bias earlier when it is not.
>
> ### 3) Comparison with recent diagnostics
>
> Thank you for raising this. We agree that the paper should better position itself relative to newer posterior diagnostics. Our method is most directly a **global hypothesis test for joint validity**, with valid $p$-values and explicit robustness to weak classifiers. Some recent methods are more local or post-hoc in nature, so they are complementary rather than direct substitutes.
>
> We have also added broader comparisons to **AutoML** and **MMDFuse**, together with matched-budget and failure-mode analyses (Please see the response to Reviwer KQwH). Recent methods (like local reshaping) are complementary post-hoc tools rather than direct substitutes. The new experiments show our main strength is stronger performance in subtle-shift regimes and substantially greater robustness when the classifier is mis-specified.
>
> ### 4) Practical adoption
>
> The right practical message is not “use any classifier”: Higher AUC is preferable because power depends strongly on ranking quality, but this is still much weaker than the standard C2ST requirement of a well-calibrated or near-Bayes-optimal classifier.
>
> Our intended practitioner guidance is therefore:
> - use **Conformal-Multiple** when budget is fixed and a drop-in replacement for C2ST is needed;
> - use **Conformal-Uniform** with moderate $m$ (e.g. $10$--$50$) when simulator draws from $p$ are cheap (in the NPE setting) and higher sensitivity to subtle errors is desired;
> - prefer continuous scores and classifiers/features with stronger AUC, since better ranking directly improves power, even though exact calibration is not required.
>
> The theory (Lemma A.3) and appendix ablations already support this tradeoff. In Appendix B, we provide an ablation over $m$ for Conformal-Uniform, which empirically confirms the power gains as $m$ increases.
>
> Thank you again for the constructive suggestions. We believe the new real-world case study, broader comparisons, and clearer practical guidance substantially strengthen the paper.

---

> > ### Author Rebuttal · Reviewer_1c8u · 2026-04-08
> >
> > I would like to thank the authors for their thorough rebuttal and the significant additional efforts made during the discussion phase.
> >
> > The authors have effectively addressed my primary concern regarding real-world validation. The addition of the gravitational lensing case study is highly commendable; it convincingly demonstrates the utility of Conformal C2ST in a practical, physics-based inverse problem and clearly shows its superiority in detecting subtle biases compared to classical C2ST and TARP. Furthermore, the explicit heuristics provided for practitioners (e.g., when to choose Conformal-Multiple vs. Conformal-Uniform) add substantial practical value to the paper.
> >
> > However, a few minor concerns remain not fully resolved. While the authors clarified the mathematical intuition behind Assumptions 2.4 and 2.6, the discussion on when and how these assumptions might realistically fail in highly complex, high-dimensional scientific spaces remains somewhat abstract. Additionally, while I understand the authors' perspective that newer local posterior reshaping methods are "complementary," a brief empirical or theoretical contrast of their distinct failure modes alongside Conformal C2ST would have provided a more complete picture of the current diagnostic landscape.
> > Given the overall high quality of the methodology, the substantial engineering effort, and the convincing empirical results, the rebuttal has successfully solidified my initial positive assessment. While the authors' responses are highly satisfactory, the preliminary nature of the adversarial robustness evaluation means I have decided to keep my score unchanged and maintain my recommendation of Weak accept (5). The paper introduces a highly significant paradigm shift for training tool-using agents, and I strongly encourage the authors to incorporate the new tables and the expanded limitations discussion into the camera-ready version.

---

> > > ### Author Response · Authors · 2026-04-08
> > >
> > > Thank you again for the thoughtful follow-up and for the positive assessment. We are especially glad that the new gravitational-lensing case study and the added practical guidance helped clarify the contribution.
> > >
> > > On the remaining points:
> > >
> > > **(1) When Assumptions 2.4 and 2.6 may fail.**
> > > We agree this should be stated more concretely and in simpler terms. Assumption 2.4 is essentially saying that, under the **true oracle density ratio**, we should *not have too many pairs of samples that are almost tied*. If many pairs are nearly tied, then the alternative is intrinsically hard for ***any*** ranking-based test to detect. In practical high-dimensional problems, dimension by itself does not make this assumption unrealistic: the key issue is not the ambient dimension, but whether the true posterior and the learned approximation are so similar, or so flat in the relevant regions, that the oracle ratio provides almost no ranking signal. A natural failure case is when both $p$ (the true distribution) and $q$ (the distribution learnt by the NPE model) are very flat over broad regions, so that the ratio changes very little and many comparisons become almost indistinguishable. Assumption 2.6 is a rare-ties condition, which can fail if the *learnt* score is heavily discretized, quantized, clipped, or nearly constant. We will revise the manuscript to make these failure modes explicit. We will also clarify that these assumptions are used only for the sharper **power/robustness** theory, not for the core finite-sample **validity** guarantee.
> > >
> > > **(2) Distinct failure modes relative to local posterior diagnostics.**
> > > We agree this comparison should be made more explicit. A limitation of our method is that it is **primarily a global test** of the NPE model over the full joint distribution: it tells us whether the learned posterior is globally consistent with the target posterior, but it does not localize *where* the mismatch occurs. In contrast, local posterior evaluation methods are designed to diagnose or correct local distortions for particular conditioning inputs of interest, for example the observed data. Thus, the methods are complementary and can fail in different ways: our method may miss a very localized discrepancy if its aggregate effect is too weak, whereas local evaluation methods can be informative about local structure without themselves providing a calibrated global hypothesis test. We will make this distinction clearer in the manuscript.
> > >
> > > **(3) Robustness experiments on real-data.**
> > >
> > > On robustness, we also want to clarify that the rebuttal already included robustness evaluations in two nontrivial high-dimensional settings: the new gravitational-lensing benchmark and the real-image/CIFAR experiment in Figure 3 (in the submitted manuscript). For convenience, we reproduce below the gravitational-lensing robustness table from our earlier response to Reviewer KQwH.
> > >
> > > **Table: Robustness at $\gamma = 1.0$ (biased samples)** — Power (%) at fixed perturbation level vs. classifier degradation $\beta$
> > >
> > > | $\beta$ | AutoML | C2ST | Conf. Mult. (ours) | Conf. Unif. (50) (ours) |
> > > |---|---:|---:|---:|---:|
> > > | 0.000 | 100 | 100 | 100 | 100 |
> > > | 0.600 | 100 | 100 | 100 | 100 |
> > > | 0.700 | 84 | 92 | 98 | **100** |
> > > | 0.800 | 44 | 40 | 74 | **86** |
> > > | 0.900 | 10 | 14 | **30** | 16 |
> > > | 0.950 | 2 | 6 | **16** | 2 |
> > >
> > > Together with Figure 3 in the submitted manuscript, this shows that the same qualitative robustness trends persist beyond the synthetic Gaussian benchmarks. We will make this empirical scope clearer in the revision.
> > >
> > > Thank you again for your constructive suggestions. We hope the above clarifies the two remaining points, and we will incorporate both clarifications into the revision to make the scope, limitations, and positioning of the method clearer. If any additional detail would be useful, we would be very happy to provide it.

---

### Official Review · Reviewer_KQwH · 2026-03-14

**Soundness:** 3
**Presentation:** 2
**Significance:** 3
**Originality:** 2
**Overall Recommendation:** 4
**Confidence:** 4

**Summary:**

The authors consider the problem of assessing whether two samples, drawn from distributions $p$ and $q$ (typically an approximation of $p$), are identically distributed. They extend previous work, the classifier two-sample test (C2ST), to the setting where the available classifier is not assumed to be near Bayes-optimal. This yields a method that relies on conformal p-values computed from an aggregation of scores, rather than a point estimate of accuracy, and is surprisingly robust to poorly calibrated classifiers. Performance is measured against C2ST, however other methods are not considered as baselines.

**Compliance With Llm Reviewing Policy:**

Affirmed.

**Final Justification:**

The authors have mostly addressed my main concerns, and I believe the paper has been improved as a result of this process. It is my opinion that soundness has been improved, and I am taking it on trust that the promised clarifications will be made to sections of the paper where it was an issue. The originality and significance has also been clarified in response to other reviewers. However, the paper still does not stand out in any of these dimensions.

For this reason, I have raised my score from a reject to a weak accept accordingly, however it is not enough for me to either champion or block the paper's acceptance.

**Key Questions For Authors:**

Can you please address the issues raised in the above section?

**Limitations:**

Yes

**Strengths And Weaknesses:**

Strengths:
- Paper is generally well communicated.
- Method is novel (to the best of my knowledge) and provides a potentially useful tool
- I personally like the method, it exploits a clever trick to generate a potentially powerful tool, and its derivation appears to be valid and performs well in the limited evaluations.

Weaknesses:
- There are a number of communication issues throughout the paper. The opening paragraph is confusing (which sets a bad tone), and should be split up. Currently it reads like a bunch of sentences stacked on top of each other, rather than a coherent paragraph. There seems to be some confusion around learning a generative model and a posterior approximation. In paragraph 2 the ordering of the product space should match the function inputs.
- Figure 3 is difficult to read.
- The biggest issue with this paper, which is likely to be a roadblock to acceptance if not addressed, is the limited evaluation. For example, comparison to AutoML-2ST, discrepancy-based methods (MMD/KSD), etc, would be useful.
- Similarly, examination of failure modes of the method (i.e., situations where other methods are more robust/perform better) would give clarity as to the utility of the method and its place in the suite of available tools.
- The paper lacks a conclusion.

---

> ### Author Rebuttal · Authors · 2026-03-31
>
> We sincerely thank you for the thoughtful review. We agree that the original submission needed broader empirical comparisons, a clearer discussion of failure modes, and several presentation fixes. We have now run the additional experiments and will revise the paper accordingly. We report multiple $\gamma$ values at fixed $\beta=0$ and multiple $\beta$ values at fixed $\gamma$. Due to rebuttal space, these are representative sub-rows from the full sweeps; the complete tables will be included in the revision / supplement. We also note that at $\gamma=0$, all methods (except for DC) controlled Type-I error near the nominal 5% level.
>
> ### 1) New comparisons to AutoML-2ST and discrepancy methods
>
> We added matched-budget comparisons against **AutoML** and **MMDFuse**. All classifier-based methods are trained on 1000 true + 1000 fake samples and evaluated over 50 replicates. MMDFuse uses the corresponding matched train+test budget.
>
> **Table 1. Power (%) at fixed classifier quality ($\beta=0$). Higher is better.**
>
> | Benchmark | $\gamma$ | AutoML | C2ST | Conf. Mult. | Conf. Unif. (50) | MMDFuse |
> |---|---:|---:|---:|---:|---:|---:|
> | Distorted variance | 0.20 | 55.33 | 56.00 | 78.00 | **85.33** | 44.00 |
> |  | 0.30 | 94.67 | 97.33 | 98.67 | **100.00** | 91.33 |
> | Perturbed mean | 0.05 | 59.33 | 58.00 | 80.67 | **87.33** | 84.67 |
> |  | 0.07 | 95.33 | 92.00 | **100.00** | **100.00** | 99.33 |
> | Perturbed variance | 0.20 | 30.67 | 23.33 | 43.33 | **50.67** | 30.67 |
> |  | 0.30 | 60.00 | 53.33 | 76.67 | **90.00** | 72.67 |
> | Tail | 0.10 | **32.67** | 15.33 | 30.00 | 31.33 | 10.00 |
> |  | 0.15 | 74.67 | 27.33 | 55.33 | **76.67** | 35.33 |
>
> These results strengthen the main empirical claim: the conformal variants are strongest in the subtle-to-moderate regime, which is the regime most relevant for posterior validation. For example, at distorted variance $\gamma=0.2$, power rises from 55.3/56.0 for AutoML/C2ST to 78.0/85.3 for Conformal-Multiple/Uniform(50); at perturbed variance $\gamma=0.3$, it rises from 60.0/53.3 to 76.7/90.0. Moreover, we see a similar trend for **high dimensional gravitational lensing** SBI experiment (please see response to Reviewer 1c8u). Both our conformal variants are more sensitive than all other baselines.
>
> These comparisons also clarify **failure modes**. Our method is not uniformly best: AutoML is slightly best on the tail benchmark at $\gamma=0.10$, and MMDFuse is competitive on perturbed mean at $\gamma=0.05$. So our claim is not universal domination, but stronger overall performance, especially on harder low-signal alternatives.
>
> ### 2) Robustness under classifier degradation
>
> Since the paper’s main claim is robustness to weak classifiers, we now show three $\beta$ values per benchmark. MMDFuse is not classifier-based, so it does not naturally participate in this experiment.
>
> **Table 2. Power (%) as classifier quality degrades.**
>
> | Benchmark | Fixed $\gamma$ | $\beta$ | AutoML | C2ST | Conf. Mult. | Conf. Unif. (50) |
> |---|---:|---:|---:|---:|---:|---:|
> | Distorted variance | 0.30 | 0.00 | 94.67 | 97.33 | 98.67 | **100.00** |
> |  |  | 0.50 | 42.00 | 0.67 | 55.33 | **56.00** |
> |  |  | 0.95 | 4.00 | 0.00 | 3.33 | **7.33** |
> | Perturbed mean | 0.07 | 0.00 | 95.33 | 92.00 | **100.00** | **100.00** |
> |  |  | 0.50 | 51.33 | 2.67 | **60.67** | **60.67** |
> |  |  | 0.95 | 23.33 | 0.00 | 32.67 | **64.00** |
> | Perturbed variance | 0.40 | 0.00 | 89.33 | 87.33 | 96.67 | **100.00** |
> |  |  | 0.50 | 60.00 | 0.00 | 78.00 | **82.00** |
> |  |  | 0.95 | 7.33 | 0.00 | 12.00 | **27.33** |
> | Tail | 0.30 | 0.00 | **100.00** | 96.67 | **100.00** | **100.00** |
> |  |  | 0.50 | 98.67 | 32.67 | 92.00 | **100.00** |
> |  |  | 0.95 | 4.67 | 0.67 | 12.67 | **34.67** |
>
> The pattern is clear: ordinary C2ST collapses quickly as the classifier degrades, while the conformal variants degrade much more gracefully. AutoML is more robust than C2ST and remains competitive on the tail benchmark, but usually trails the conformal variants.
>
> ### 3) Presentation fixes
>
> We also agree with the presentation concerns and will revise the paper accordingly: we will rewrite the opening paragraph, fix the product-space notation mismatch, enlarge and simplify Figure 3, and revise the title/message so it is clearer that the contribution is Conformal C2ST, not ordinary C2ST.
>
> Overall, your review helped us strengthen the paper in the most important direction. The revised version will include broader baselines, a clearer discussion of failure modes, and a stronger empirical case for our message: the **main strength of conformal C2ST** is not universal domination, but (i) better sensitivity subtle posterior errors, and (ii) much greater robustness when the classifier is weak or mis-specified.

---

> > ### Author Rebuttal · Reviewer_KQwH · 2026-04-04
> >
> > Thanks for your rebuttal to my review and addressing a number of the claims.
> >
> > I understand that the character lengths of responses are frustrating and require certain things to be truncated. However, the corrected/clarified/rewritten components, as well as your experimental details are not clear to me at this stage. Is it possible to clarify these in your next response, and post a link to an anonymized GitHub so I can look at the code myself? Tables and figures can also be linked if such a repository is set up. If the additional experiments are synthetic then comparisons for real data would be *highly* valuable, as this is an empirically driven field. I do appreciate that it is not a requirement for a method to perform uniformly 'better' in order for it to be valuable.

---

> > > ### Author Response · Authors · 2026-04-06
> > >
> > > Thank you for the follow-up. We apologize for the confusion due to the compressed explanation. We clarify the setup below and include exact implementation details with even more fine-grained results tables for maximum transparency.
> > > # Setup
> > > Our synthetic benchmarks (Tables 1,2 above) use the Appendix B perturbations (distorted variance, perturbed mean/variance/tail). There are two independent parameters:
> > > - **$\gamma$** controls posterior misspecification ($\gamma=0\Rightarrow q=p$; larger $\gamma$ means a more distorted posterior and an easier testing problem).
> > > - **$\beta$** controls classifier degradation ($\beta=0$ is the fully trained classifier; larger $\beta$ linearly interpolates the classifier toward random initialization, making it weaker).
> > >
> > > Hence the fixed-$\beta=0$ tables (Tables 1,3) measure **sensitivity** (better = higher power at smaller $\gamma$), while the fixed-$\gamma$ tables (Tables 2,4) measure **robustness to weak classifiers** (better = higher power for larger $\beta$).
> > >
> > > # Scientific benchmark: gravitational lensing
> > > We agree with the reviewer that simulated experiments alone are not enough. Our paper already includes a real-image distribution-shift setting in Figure 3. To better address the reviewer’s request, we also added a **non-synthetic gravitational-lensing benchmark** in the NPE context, following the TARP paper (Lemos et al., 2023, cf. **Appendix E**).
> > >
> > > **Setup.** They fit a prior over latent source images $\theta \in \mathbb{R}^{256}$ (PROBES dataset) and get posterior samples from the reverse SDE of a score-based diffusion using conditional score decomposition. Unlike Lemos et al., who only compare exact vs. biased endpoints, we introduce a continuous interpolated score to measure how early bias is detected:$$s_\gamma(\theta_t\mid x) = (1-\gamma)s_{true}(\theta_t\mid x) + \gamma s_{biased}(\theta_t\mid x)$$Here, $\gamma=0$ yields the exact posterior sampler, and $\gamma=1$ is fully biased.
> > >
> > > **Budget.** In each replicate we use $n_{\text{true}}=n_{\text{fake}}=2000$, one posterior sample per $x$ for training, and report rejection rates over 50 independent replicated test sets. MMDFuse again uses the matched train+test budget.
> > >
> > > **Results.** Let **CM=Conformal Multiple**, **CU(50)=Conformal Uniform (50)**.
> > >
> > > Table 3: **Sensitivity at $\beta=0$**: Power (%) at fixed classifier quality vs perturbation level $\gamma$.
> > > | $\gamma$ | AutoML | C2ST | CM (ours) | CU(50) (ours) | DC | MMDFuse | SBC | TARP |
> > > |---|---:|---:|---:|---:|---:|---:|---:|---:|
> > > |0.00|4|2|2|5|10|8|10|4|
> > > |0.40|10|8|8|8|**16**|2|6|4|
> > > |0.50|6|14|**24**|16|22|12|4|2|
> > > |0.60|28|28|50|**66**|46|14|6|6|
> > > |0.70|70|76|92|**98**|52|18|6|6|
> > > |0.80|**100**|**100**|**100**|**100**|66|24|8|6|
> > > |0.90|**100**|**100**|**100**|**100**|80|24|2|2|
> > > |1.00|**100**|**100**|**100**|**100**|98|28|6|24|
> > >
> > > Table 4: **Robustness at $\gamma=1.0$**: Power (%) at fixed perturbation level vs classifier degradation $\beta$.
> > > |$\beta$|AutoML|C2ST|CM (ours)|CU (50) (ours)|
> > > |---|---:|---:|---:|---:|
> > > |0.000|100|100|100|100|
> > > |0.600|100|100|100|100|
> > > |0.700|84|92|98|**100**|
> > > |0.800|44|40|74|**86**|
> > > |0.900|10|14|**30**|16|
> > > |0.950|2|6|**16**|2|
> > > # Conclusion
> > > We deliberately introduce the simulated Gaussian benchmarks- the aim is not realism but controlled stress-testing where the ground truth posterior is known. Even in those simple settings, our methods generally outperform all baselines. Our results on the real-world gravitational-lensing benchmark (and Figure 3) show that the same trends also hold in real scientific applications, establishing the practical applicability of our methods.
> > > - In the **fixed-$\beta$ sensitivity table**, the conformal variants usually detect discrepancies at smaller $\gamma$ than C2ST/AutoML/MMDFuse/TARP/SBC, i.e. our methods are more sensitive in the subtle-shift regime
> > > - In the fixed-$\gamma$ robustness tables, **Conformal-Multiple already substantially outperforms ordinary C2ST and AutoML under matched budget**, and in some cases even Conformal-Uniform(50), so the gain is not coming only from extra samples from $p$
> > > - **Conformal-Uniform** is strongest (overall) due to larger effective calibration and fresh, independent calibration sets per test point. (Under severe degradation, all methods eventually approach the nominal type-I-error level)
> > > # References
> > > The baseline methods are directly taken from their official code or the standard sbi repository (https://github.com/sbi-dev/sbi).
> > > 1. SBC: SBI repo
> > > 2. TARP: github.com/Ciela-Institute/tarp
> > > 3. AutoML: github.com/jmkuebler/autoML-TST-paper
> > > 4. C2ST: SBI repo
> > > 5. DC: github.com/yao-yl/DiscCalibration
> > > 6. MMDFuse: github.com/antoninschrab/mmdfuse
> > > 7. **Ours:** We are unable to share an anonymized repository during rebuttal, but will happily provide the code upon acceptance. The implementation strictly follows the methodology described in the submitted manuscript with the budgets listed above.
> > >
> > > We hope this addresses your concerns about both the setup and the role of the added experiments.

---

### Decision · Program_Chairs · 2026-04-30

**Decision:**

Accept (regular)

**Comment:**

An interesting paper, with strengths and weaknesses.

I appreciated the reply the authors did to KQwH. After the last experiments the authors provided, I am happy to tip my evaluation of the paper towards the author's argument, *but* they absolutely have to update the draft to include the last experiment provided at rebuttal time. Without any doubt, it will contribute to improve the paper's quality. I also appreciated the reply done to 1c8u, in particular when assumptions may fail. Again, it is *very* important to push this sort of consideration to the main body because assumptions can be a bit obscure and understanding when they fail or not can help provide an idea of the importance of those.

Finally, I appreciated the detail the authors put in comparing their approach to that of Hu and Lei (t9Vt). The questions of the reviewer were legitimate, so it would be a good idea to reinforce the differences between the approach the authors propose and the SOTA by adding a substantial subsection on this. As it stands, the draft gives an impression of "diluting" those comparisons in diverse sections and does not give enough credit to the contribution of the authors.